# The Cell Transformation Assay: A Historical Assessment of Current Knowledge of Applications in an Integrated Approach to Testing and Assessment for Non-Genotoxic Carcinogens

**DOI:** 10.3390/ijms24065659

**Published:** 2023-03-16

**Authors:** Annamaria Colacci, Raffaella Corvi, Kyomi Ohmori, Martin Paparella, Stefania Serra, Iris Da Rocha Carrico, Paule Vasseur, Miriam Naomi Jacobs

**Affiliations:** 1Agency for Prevention, Environment and Energy, Emilia-Romagna (Arpae), Via Po 5, I-40139 Bologna, Italy; 2European Commission, Joint Research Centre (JRC), I-21027 Ispra, Italy; 3Chemical Division, Kanagawa Prefectural Institute of Public Health, Chigasaki 253-0087, Japan; 4Research Initiatives and Promotion Organization, Yokohama National University, Yokohama 240-8501, Japan; 5Division of Medical Biochemistry, Biocenter, Medical University of Innsbruck, A-6020 Innbruck, Austria; 6Universite de Lorraine, CNRS UMR 7360 LIEC, Laboratoire Interdisciplinaire des Environnements Continentaux, 57070 Metz, France; 7Radiation, Chemical and Environmental Hazards, UK Health Security Agency, Harwell Science and Innovation Campus, Chilton OX11 0RQ, UK

**Keywords:** rodent cell transformation assay, rodent cancer bioassay, transcriptomics, transformics chemical-induced transformation, mechanistic understanding, chemical-induced transformation, enrichment analysis, tumor microenvironment, in vitro oncotransformation, carcinogenesis

## Abstract

The history of the development of the cell transformation assays (CTAs) is described, providing an overview of in vitro cell transformation from its origin to the new transcriptomic-based CTAs. Application of this knowledge is utilized to address how the different types of CTAs, variously addressing initiation and promotion, can be included on a mechanistic basis within the integrated approach to testing and assessment (IATA) for non-genotoxic carcinogens. Building upon assay assessments targeting the key events in the IATA, we identify how the different CTA models can appropriately fit, following preceding steps in the IATA. The preceding steps are the prescreening transcriptomic approaches, and assessment within the earlier key events of inflammation, immune disruption, mitotic signaling and cell injury. The CTA models address the later key events of (sustained) proliferation and change in morphology leading to tumor formation. The complementary key biomarkers with respect to the precursor key events and respective CTAs are mapped, providing a structured mechanistic approach to represent the complexity of the (non-genotoxic) carcinogenesis process, and specifically their capacity to identify non-genotoxic carcinogenic chemicals in a human relevant IATA.

## 1. Introduction

Carcinogenesis is a multistep process, one that has been conventionally separated into three subsequent steps, including initiation, promotion and progression. According to this view, cancer is initiated by genotoxic damage within a single cell, the initiated cell, whose proliferation sustains the origin of tumor, then progresses through the accumulation of further genetic changes [1,2].

Here we apply the consensus agreed definitions of genotoxic and non-genotoxic carcinogens, as relevant for regulatory purposes: ‘The induction of cancer involves the accumulation of genomic alterations, which can be induced directly or indirectly. Carcinogens have conventionally been divided into two categories according to their presumed mode of action, genotoxic carcinogens and non-genotoxic carcinogens. A genotoxic carcinogen has the potential to induce cancer by interacting directly with DNA and/or the cellular apparatus involved in the preservation of the integrity of the genome. A non-genotoxic carcinogen has the potential to induce cancer without interacting directly with either DNA or the cellular apparatus involved in the preservation of the integrity of the genome’ [3].

The somatic mutation theory of carcinogenesis was formulated by Theodor Boveri in 1914 and published in 1929 [1]. Since then, it has provided the rationale for the use of mutagenesis tests to demonstrate a chemical’s ability to induce mutations and potentially initiate the carcinogenic process. Whilst this approach allowed reduction in the use of the rodent carcinogenicity bioassay (RCB), due to an extensive use of cheaper, shorter and easier in vitro mutagenicity tests, it has in some way led to the misconception in the use of term ‘carcinogen’, which, for a long time, has been considered equivalent to genotoxic chemicals, and ‘non-carcinogen’, and often referred to chemicals that do not induce mutations. This assumption created an assessment gap in the identification of chemicals that can initiate and sustain the entire process of carcinogenesis through non-genotoxic mechanisms, unless the RCB is used to identify them, as discussed in earlier related publications [3,4].

The standard RCB requires an extensive use of animals. Apart from animal welfare considerations, it shows several limitations, particularly due to the high costs, the prolonged duration (2 years), and the scarce mechanistic information generated, which can make it difficult to extrapolate for human relevance [5,6]. Further, international work is ongoing with the aim to review the uncertainty and complexity of the RCB-based assessments. This is contributing to revisiting RCB reference data evaluation and is also improving the definition of acceptable performance of in vitro approaches [4,5,6].

It was in this context that the international regulatory community agreed on the need to develop an integrated approach to testing and assessment for non-genotoxic carcinogens (IATA-NGTxC) [3,4] and an OECD expert group was subsequently formed.

In 2020 the expert group published a consensus paper describing the overarching IATA developed, with the molecular initiating events of cellular metabolism and receptor interactions, followed by the early key events of inflammation and immune dysfunction, mitotic signaling, cell injury, leading to (sustained) proliferation then morphological transformation leading to tumor formation [3].

Amongst the in vitro tests that are under consideration for inclusion in the IATA-NGTxC, the cell transformation assay (CTA) was the test method that triggered the initiation of the OECD NGTxC IATA activity, and it is the CTA that is the focus of this historical review. The other key events are considered in related papers, including, for example, cell proliferation (Strupp et al. paper in preparation). The CTA was first developed 60 years ago as a tool to explore the multistep carcinogenesis (initiation and promotion) process and soon became a model to study chemical carcinogenesis.

Collectively, the experimental evidence described herein shows that cellular and molecular processes involved in in vitro cell transformation seem to resemble those sustaining in vivo carcinogenesis, and these processes occur as a result of comprehensive cellular responses to direct and indirect damage to DNA.

The CTA can thus provide some additional critical information to support other existing tests for assessing carcinogenic potential [7,8,9,10,11,12,13]. The CTA measures the morphological transformation of cells, as transformed colonies or malignant foci derived from a single cell. It is considered to involve a multistage process that closely models some stages of in vivo carcinogenesis. However, further mechanistic understanding was needed to better ascertain how the CTAs actually resemble in vivo carcinogenesis, with an understanding of human relevance, in order to understand how it might be integrated into the IATA-NGTxC. Subsequently, the discussion within the IATA-NGTxC expert group boosted research on the molecular events sustaining the process of oncotransformation in vitro and revitalized the interest towards the CTA.

This paper reports upon the (unpublished) CTA data evaluations conducted under the auspices of the OECD, when discussing the possible adoption of the CTA as a Test Guideline, and reflects the ongoing CTA development discussions. It provides an overview of in vitro cell transformation, from its origins to the new transcriptomic-based CTAs. The aim is to address still open questions regarding the feasibility of alternative approaches, in particular the CTA, to represent the complexity of the carcinogenesis process, and specifically the capacity to identify non-genotoxic carcinogenic chemical hazards, for improved human health protection.

### A Historical Perspective on Cell Transformation Assay Models

The first experiments demonstrating that normal cells can be transformed into tumor cells in vitro date back to the 1950s, when several viruses, including Rous sarcoma, avian myeloblastosis and SE polyoma viruses, were found to be able to induce fibroblastic transformation in susceptible cells [14,15,16,17]. Subsequent experiments demonstrated that in vitro cell transformation was also induced by chemicals and X-ray irradiation. These pioneering experiments, set up to provide an in vitro model to study oncotransformation, gave evidence for the inherited capability of transformed cells to evade contact inhibition, growing in vitro in a random arrangement, forming tumors when injected into suitable hosts.

The first method to study the chemically induced transformation in vitro was introduced in the early 190s by using Syrian hamster embryo (SHE) cells [18], which have previously been successfully employed to study virus-induced transformation [17] (Figure 1). SHE cells are normal diploid, metabolically competent primary cells obtained from 13-day old embryos, a developmental stage prone to morphological transformation upon exposure to carcinogenic agents. An early, extensive characterization of SHE cells, performed to support their use as a model of carcinogenesis in vitro, demonstrated that cultivated embryo cells initially included multiple types of cells at different stages of differentiation, which became uniform after the fifth or sixth passage. At this passage stage, they showed a growth pattern typical of fibroblast-like cells [19]. Even if the mitotic rate of cultured cells decreased with the time and cells acquired a different morphology, hamster embryo cells did not give rise to established cells, thus showing a behavior typical of most fibroblast-like cells of human origin [19].

Under standard culture conditions, SHE cells are genetically stable, showing a low rate of spontaneous transformation and a modest plating efficiency, which increases if cells are allowed to grow in a semisolid medium [19,20]. When treated with transforming agents, SHE cells give origin to aberrant colonies of fusiform disoriented cells derived from single parental cells and characterized by a random arrangement of cells, which criss-cross at the edge of transformed colony and have a tendency to pile up [19,20,21].

One of the main advantages of using the SHE model is represented by the possibility of studying the very early steps of the process leading to malignancy, since SHE cells are not established cells, i.e., cells that have acquired the ability to proliferate, and nor are they a cell line, which is a permanently established cell culture. Moreover, as primary cells, SHE cells retain the ability to biotransform a wide range of xenobiotics as evidenced by studies with substances requiring metabolic activation [22].

However, established cells can offer additional advantages, such as cellular uniformity and high plating efficiency.

The BALB/c 3T3 model was the first CTA developed that was based on a cell line. This was conducted by using the clone A-31 from BALB/c mouse strain [23] (Figure 1).

Initially used to study virus-induced cell transformation, it was then employed to examine tumor promotion in vitro [24].

BALB/c3T3 are embryonic mouse fibroblasts, which undergo transformation following chemical treatment, with cells escaping contact-inhibition and piling up randomly. The endpoint of transformation is represented by the formation of foci of altered multi-layered, disorganized, anchorage-independent cells forming on a monolayer background of confluent contact-inhibited cells. The transformed cells from malignant foci are tumorigenic and metastatic when injected into suitable host animals and acquire invasive properties in vitro [25,26,27]. Contrary to other established cell lines [28], BALB/c 3T3 cells still retain enough metabolic activity to support both phase-1 and phase-2 metabolic activation of most carcinogens [29], which is valuable considering that the majority of carcinogens require bioactivation.

BALB/c 3T3 cells belong to the so-called ‘3T3 cell lines’, which include several clones of primary mouse embryonic fibroblasts. The 3T3 term refers to the cell propagation protocol, according to which cells are seeded at the critical density of 3 × 10^5^ cells/plate and split (transferred) every 3 days. Cells are then maintained in culture until they reach a stable growth rate. 3T3 cell lines were initially derived from Swiss albino mouse (Swiss-3T3 cells, NIH-3T3 cells) in 1962 [30]. Other cell lines were obtained as A31 subclones (clone A31-714, clone A31-1, clone A31-13) [31]. A further clone of BALB/c 3T3, clone A31-1-1, was established in 1980, following a procedure of subcloning of BALB/c 3T3 A31 [31]. The BALB/c 3T3 A31-1-1 clone was selected for its relative high susceptibility to transforming agents. However, a subsequent genotyping characterization revealed that the BALB/3T3 A31-1-1 cell line, was misidentified and actually originated from Swiss mouse [32]. This could explain differences in sensitivity to chemical-induced transformation, as well as the differences in metabolic competence, due to genetic variations in the metabolism controls between mouse strains [29,31]. Three other clones were obtained by transfecting the clone 3T3 A31-1-1 with (onco)genes, which included Bhas 42 cells transfected with an activated v-H-Ras, 1-1ras1000 cells transfected with human activated c-Ha-ras, and 1-1Src cells transfected with the avian v-src [33,34] (Figure 2). A 3T3-like cell line was also established from SHE cells, showing properties similar to those of mouse 3T3 cells [35]. However, this cell line was refractory to chemical-induced transformation [21].

The mouse C3H 10T1/2 cell line represents another CTA model. C3H 10T1/2 cells were isolated in 1972 from the C3H mouse and established by replating 0.5 × 10^5^ cells/60 mm plate every 10 days [36] (Figure 1). The chemical treatment of C3H 10T1/2 cells induces the formation of three types of foci, two of which (Type-II and Type-III) show a complete malignant phenotype.

Of all these cellular models developed in the past, three are currently most used for testing chemically induced transformation in vitro: they are the SHE model, the BALB/c 3T3 model and the Bhas 42 CTA. While the Bhas 42 has been specifically developed to test chemicals acting as initiating agents (including genotoxic) or promoters (including non-genotoxic) in the multistep carcinogenesis process, the SHE, BALB/c 3T3 and C3H 10T1/2 models have also been initially utilized with the same objective.

Indeed, these three models may also detect chemicals acting as initiators or promoters, using a sequential treatment as conducted during the first decades of their development [37]. However, the OECD guidance document on the SHE CTA (2015) [22] did not focus on an initiation-promotion protocol, retaining only a one-step protocol.

Bhas 42 cells have been established by Sasaki et al. from BALB/c 3T3 A31-1-1 cells through the transfection with a plasmid containing v-Ha-ras gene [38].

Untransformed Bhas 42 cells grow to confluence forming a contact-inhibited monolayer. They are not tumorigenic upon transplantation in vivo. After exposure to carcinogenic agents, from the sub-confluent cell density, Bhas 42 cells form transformed foci, rising from morphologically altered cells, which acquire the ability to invade the surrounding non-transformed contact-inhibited monolayer. Ohmori et al. developed a tumor-promotion test method for detecting NGTxC [39], the highly robust protocol included the addition of preculture conditions and was optimised in an inter-laboratory collaborative study involving 14 laboratories from the Japanese non-genotoxic carcinogen study group [40].

Since Bhas 42 cells express an activated v-Ha-ras oncogene, they are regarded as already initiated cells, according to the two-stage paradigm of genotoxic carcinogenesis [33,41]. A high-throughput version of the assay has also been established [42].

The original CTA protocols have undergone significant changes and amendments over the years to improve the sensitivity and specificity of the method. However, to address whether the available protocols were sufficiently standardized to support their routine use for regulatory purposes, the European Centre for the Validation of Alternative Methods (ECVAM) coordinated a study to address issues of CTA protocols standardization, transferability and reproducibility in 1998 [43].

The SHE CTA, which was originally performed at a neutral pH (7.2–7.3), was then carried out in an acid environment (pH 6.7) to increase the sensitivity of the assay and cells’ responsiveness to different chemicals [44].

The BALB/c 3T3 CTA original protocol, which requires seeding cells at a density of 1 × 10^4^ cells/plate and allowing cells to grow 24 h before treatment, was modified by increasing cell density and changing treatment schedule, to reduce the rate of cytotoxicity and increase specificity of the method of the standard model (one step) [45,46].

Two assay modifications were introduced to increase the yield of malignant clones in C3H 10T1/2 cells, including the S9 liver fraction supplementation as an exogenous metabolic system, and a Level-II amplification, based on reseeding treated cells to allow the selection of quiescent transformed foci [24,47].

Considerable efforts were also made to improve the capacity to discriminate between non-transformed and transformed clones.

Indeed, all CTA models provide an easily detectable endpoint of oncotransformation which is based upon morphological characteristics and can be used to anchor the test chemical exposure to the acquisition of the malignant phenotype.

However, subjectivity in identifying morphologically transformed foci or colonies has often been indicated as one of the main limitations of the CTAs [48]. To overcome this limitation, photo-catalogues were provided as a visual aid for the identification and the scoring of foci and colonies in the conduct of the assay in 3T3 and SHE cells, respectively [49,50]. Automated imaging tools for the scoring of 3T3 foci were also proposed, mostly to support naïve laboratories to set up CTA protocols [51,52], and a method based on vibrational spectroscopy was proposed to score SHE colonies [20]. More recently, a basic computational model using convolutional neural network (CNN) deep learning for the purpose of automatic focus determination in the Bhas 42 CTA was developed [53].

Despite efforts to improve the method and encouraging feedback from studies that have explored the transforming ability of a huge number of chemicals with different mode and mechanisms of action, the CTA has not been considered sufficiently well understood to predict non-genotoxic carcinogenicity when used on its own. It was therefore not adopted as an OECD Test Guideline. Instead, two guidance documents were issued by the OECD to support the use of CTA based on SHE cells and Bhas 42 cells respectively [22,54]. Additionally, a protocol for BALB/c 3T3 CTA has been recommended by ECVAM on the basis of a pre-validation study [52]. However, following a discussion within the OECD Working Group of National Coordinators of the OECD Test Guidelines Programme in 2014, the international regulatory community agreed to consider how the CTA, along with other relevant test methods, could be included within the IATA-NGTxC [3,4]. Figure 1 provides a timeline showing the milestones in the development of the CTA. Figure 2 provides a diagrammatic overview of the origin of the established 3T3 mouse embryonic fibroblasts.

All cell lines originate from two different mouse strains: Balb/c inbred mouse and Swiss albino outbred mouse. Only the 3T3 clone A31 from BALB/c mouse, the clone A31-1-1 from Swiss outbred mouse and the Bhas 42 clone obtained from A31-1-1 transfection are currently in use for CTA purposes.

## 2. Cellular and Molecular Mechanisms of Cell Transformation and Their Significance for In Vivo Carcinogenesis

### 2.1. The Role of Fibroblasts in Cancer

Historically, cells for in vitro transformation studies were usually obtained from animal embryos, especially chicken, mouse, rat and hamster. Cells from primary tissue gave origin to rapidly dividing fusiform cells, dominating cell cultures, which were first considered and later proved to be fibroblasts [55]. This observation was initially justified by the fact that malignant tumors can arise from cells capable of actively dividing, including cells of mesenchymal origin. It was only much later that the inherent plasticity and resiliency of fibroblasts, and their key role in maintaining tissue integrity, was recognized. Indeed, normal fibroblasts were initially shown to be capable of inhibiting the growth of cancer cells in vitro (neighbor suppression) [56], whilst the crucial role of fibroblasts in tumor progression and metastasis has only emerged in the last few years [57].

Fibroblasts are usually quiescent cells that become activated through epigenetic mechanisms, in response to stress, mechanical changes and tissue damage signals. Fibroblast activation, which is a reversible process under physiological conditions, leads to the production of extracellular matrix (ECM) and modulation of inflammation [58]. Activated fibroblasts also sustain proliferation and differentiation of epithelial cells. These processes are regulated by chemokines and cytokines produced by activated fibroblasts in order to maintain the communication with other mesenchymal, epithelial and immune cells [58]. All these functions are used and enhanced in fibroblasts recruited in the tumor microenvironment [57,59]. Even if the transition process from quiescent and/or activated fibroblasts to cancer-associated fibroblasts (CAFs) is not completely elucidated, the pro-tumorigenic function of dysregulated quiescent fibroblasts and the primary role of CAFs in the evolution and progression of cancer are both well recognized [57,58]. Therefore, fibroblasts are considered to be a good cellular model to understand key molecular events that directly support carcinogenesis and, as such, the current models for cell transformation in vitro are a useful tool to elucidate these events.

Furthermore, colonies of epithelial cells and epithelial type disoriented cells in the CTA have been observed [50]. The three CTA models currently used include primary cells of fibroblastic origin (SHE), established mouse fibroblasts (BALB/c 3T3) and oncogene-transfected fibroblasts (Bhas 42).

The capacity of fibroblasts to grow in vitro is species-specific. Human fibroblasts have a higher proliferative potential and higher resistance to oncogene-transformation than mouse cells [60]. Hamster fibroblasts seem to display similar behavior to that of human fibroblasts [55,60].

When mammalian fibroblasts are placed in culture, they may replicate at a higher rate than they do in the intact organism, depending upon the components of the culture medium. However, this proliferative rate declines over time and eventually ceases.

Mouse embryonic fibroblasts (MEFs), placed in the 3T3 protocol, stop dividing after 15–30 generations and cells show high levels of negative regulators of cell cycle, including p16, p21, p19 and p53 [60]. Rare immortal variants can emerge and originate from 3T3 established cell lines, which acquire the ability to proliferate much faster at low density than original MEFs [60,61]. Immortality is accompanied by genomic variations. Most immortalized cells are hypotetraploid and contain mutant p53, while others are pseudodiploid and characterized by INK4a/Arf locus (p16) deletion [55,60]. Pseudodiploid cells can easily revert to diploid cells, restarting the mitotic clock related to the senescence process [55]. Dysregulation of p53 seems to be strictly connected to the acquisition of polyploidy and cell fate to immortalization seems to be the result of disabling ARF-Mdm-p53 checkpoint [60].

The same rare immortal cells can be selected by the treatment of normal diploid cells, such as SHE cells, with carcinogenic chemicals [55]. Not all immortalized cells show p53 dysregulation or p16 epigenetic modifications, confirming that immortalization and transformation of primary cells are insufficient for senescence bypassing [61].

Immortalized cells may sometimes progress to anchorage independence and malignancy [55]. Progression to malignancy can be triggered by maintaining cells in culture, allowing cell growth and replication, or by exposing cells to further treatments.

Immortality is therefore a necessary but insufficient condition to drive forward malignant transformation.

This reflects the role and fate of normal fibroblasts and CAFs in tumor progression in humans. Primary human fibroblasts are resistant to immortalization in vitro. Even when p53 or other oncosuppressors such as Rb are dysregulated, human cells maintain the senescence control much longer than MEFs. However, when looking at the events at the tissue level, the loss of TP53 gene in normal human fibroblasts leads to the formation of CAFs, and these play a key role in the process of stromatogenesis, which, in turn, supports the tumorigenesis process. Moreover, CAFs can reprogram p53, altering its role of oncosuppressor by gaining specific mutant ability to sustain tumor formation and progression [62]. Therefore, malignancy is the combined consequence arising from p53 loss together with a functional gain orchestrated by fibroblasts. Further molecular steps are then needed for the acquisition of a fully malignant phenotype.

### 2.2. Multistep Oncotransformation Models

Early studies have provided the evidence for the main molecular key events supporting the process of oncotransformation in vitro, showing (surprising) overlap with the key molecular events in human cancer (the information from these studies is summarized in Figure 3).

The multistep genetic model of human colon carcinogenesis, as described by Fearon and Vogelstein in 1990 [63], is still considered the paradigmatic example for solid tumors. The description of events at molecular, cellular and tissue level in colorectal tumorigenesis is usually used to understand tumor progression associated with both genotoxic and non-genotoxic carcinogenic mechanisms. This model also provides evidence for the critical number of genetic hits (mutations) required for the progression of solid tumors [1,64]. On this basis, the colon model can be used as the epitome of the multistep carcinogenesis process.

Colon carcinogenesis can originate through three different pathways, including the chromosomal instability (CIN) pathway, the sporadic microsatellite pathway (MSI) and the pure MSI pathway from germline mutations in a DNA mismatch repair (MMR) gene, and the CpG methylation pathway. The genetic model proposed by Fearon and Vogelstein [63] is representative of the CIN pathway, which accounts for up to 85% sporadic colon cancers and for hereditary colon cancer in familiar adenomatous polyposis (FAP) patients [65]. The CIN pathway in colon cancer is characterized by mutation in the APC oncosuppressor gene or deletion of chromosome 5q, containing APC, mutation of K-RAS, deletion of chromosome 18q and deletion of chromosome 17p, containing TP53. These alterations lead to the loss of gene function and dysregulation of pathways essential to the initiation and progression of colon cancer, have been described elsewhere and are summarized in Figure 3.

The paradigmatic example of human colorectal cancer is represented, considering two pathways sustained by chromosomal instability (CIN): the sporadic colorectal adenocarcinoma and the inflammation-associated colorectal model. Only chromosomal changes are reported, including mutations, loss of heterozygosis (LOH) or gain of function. Each molecular change is then associated with dysregulation of key signal pathways, as described elsewhere for both human cancer [66] and 3T3 CTAs [67,68,69].

Each of these molecular changes have been reported to be highly associated with human cancer, even if only a minority of colorectal cancers are characterized by the complete set of genetic abnormalities [66]. A similar model was proposed by Rhodes and Campbell for inflammation-associated colorectal cancer, including the same key molecular events, whose sequence, however, in the multistep process, is inverted [70] (Figure 3).

Despite the discrepancies between human and rodent carcinogenesis processes [6], there is a certain degree of similarity regarding the molecular changes required for tumor progression. This is particularly true for colon carcinogenesis. The role of APC and the related beta-catenin loss of function has been demonstrated in mouse [71,72]. Thus, in this regard, the Apc-deficient mouse model (ApcMin/+ mouse, Apc mutant mouse) represents a human relevant model for studying chemical carcinogenesis in vivo [73]. Point mutations in K-RAS have been reported as the preferential endpoint in intestinal adenocarcinoma in mouse induced by chemical carcinogens [71]. Tumor progression in rodents is also marked by loss of heterozygosity (LOH) in several chromosomes, but it is a much more accelerated process than in humans, due to interspecies differences in the control of senescence [71].

With the advent of new molecular technologies and with a better understanding of the key events in the multistep carcinogenesis process, it has become possible to explore the complete pattern of molecular modifications characterizing the oncotransformation in vitro. This gives greater evidential support to the early observations on the role of genes related to cancer and clarifies some (apparent) discrepancies and incongruities from early studies of cellular and molecular mechanisms of cell transformation [61,67,68,69,74].

On the basis of evidence acquired in recent years, it is possible to depict oncotransformation in vitro as a multistep process, which for many, but not all cancers, is initiated by dysregulation of p53, sustained by the activation of RAS and progressed to malignancy through a cascade of events related to the dysregulation of several gene pathways (Figure 3).

The most recent reports support the hypothesis that the three currently used models for the CTA may each be representative of different steps of carcinogenesis.

As already described, SHE cells are more resistant to immortalization than 3T3 cells, rare immortal variants from SHE cells in culture are pseudodiploid and can revert to a normal phenotype [19]. Chemical-induced transformation, and immortalization supported by p53 mutations, are not always sufficient to guarantee senescence bypass in SHE cells [61]. A combination of two steps, p53 mutation and p16 transcriptional gene silencing, are required to evade senescence and confer fully malignant phenotype to SHE transformed cells [61]. P16 silencing is strictly related to p16 gene promoter hypermethylation [61]. The P16 gene (p16INK4a or CDKN2A or INK4/ARF locus) is located on 9p21 chromosome and acts as a tumor-suppressor gene. It encodes for a 16Kd protein that inhibits cyclin D kinases CDK4 and CDK6, which, in turn, regulate cell progression through the G1 phase of cell cycle (Figure 4). P16 silencing is considered the second most frequent change in human cancer [75,76]. The state of methylation of p16 has high prognostic value, especially in colon cancer, where p16 inactivation is associated with a high level of K-RAS and BRAF mutations. CDKN2A is one of the first genes recognized in the alternative pathway leading to colorectal cancer, represented by the CpG island methylator phenotype (CIMP) [77,78]. CIMP is a subset of colon cancers characterized by epigenetic instability, following hypermethylation of oncosuppressor genes promoters. CIMP is considered an event that appears early in the process of colon carcinogenesis and sustains cancer progression. At the tissue level, CIMP is related to the serrated pathway of colorectal tumorigenesis rather than to the adenocarcinoma pathway [79]. More generally, CDKN2A is considered a biomarker in human carcinogenesis, and its methylation is the major mechanism by which cells can acquire an advantage in progression, in most human cancers [80].

The inactivation of p16 via the methylation of its promoter has been described as a possible mechanism of chemical carcinogenesis via non-genotoxic events as reviewed in a sister publication in this special issue, Desaulniers et al. [74]. As previously reported, the control of cell-cycle progression is species-specific, with CDKN2A more human- than mouse-specific [74]. This confirms that SHE cells are more representative of human cell behavior than mouse cells and means that CDKN2A hypermethylation and p16 silencing are suitable biomarkers to understand the early events following human chemical exposure, in relation to tumor progression. A schematic representation clarifying the concordance between CIMP-sustained serrated pathway of colorectal cancer, and SHE cells immortalization and formation of aberrant tumorigenic colonies, following chemical exposure, is shown in Figure 4.

Molecular steps leading to the acquisition of malignancy have been elucidated in 3T3 cells used to study cell transformation in vitro. Established 3T3 cells are all characterized by p53 deficiency [27,81] as a requirement for immortalization. However, non-transformed established 3T3 cells also carry mutant RAS. RAS plays an elusive role in 3T3 cell transformation. RAS is able to induce transformation in MEFs, supporting focus formation and increasing saturation density without inducing serum-independent growth, typical of a fully malignant phenotype [82]. Indeed, RAS-induced transformation is reversible in the absence of immortalization, showing that fibroblast immortality is a pre-requisite for transformation, and/or further molecular-genetic changes [82]. Early studies did not provide strong evidence of the role of specific chemical-induced mutations in TP53 and RAS in the acquisition of a malignant phenotype [83]. It was only later that it was possible to define the specificity of certain, but not all, RAS mutations in cancer progression [84]. However, both TP53 and RAS undergo amplification as a result of progression to malignancy [27].

The role of p16 in inducing senescence by-pass is shown for both human cancer and in vitro cell transformation.

### 2.3. Molecular Signatures in In Vitro Cell Oncotransformation

The application of transcriptomics tools has made it possible to elucidate the molecular events that mark the steps to cell transformation in vitro [67,68,69,85]. Results from these studies, performed using the three currently most used CTA models, show an excellent level of concordance and resemblance of the human carcinogenesis process (Figure 3 and Figure 4).

Cytoskeleton remodeling is one of the first events highlighted by the analysis of gene modulation induced by different chemicals, recognized as carcinogens via genotoxic and non-genotoxic mechanisms [67,68,69,85]. The cytoskeleton is involved in the innate response to danger signals from xenobiotics, leading to an adaptive response [86]. However, the cytoskeleton also plays a key role in the acquisition of a malignant phenotype in human cancer, marking the passage from an adaptive to maladaptive response and supporting the process of invasion and metastasis [87]. Several genes orchestrate the dual role of the cytoskeleton. Most of these genes and related biological pathways have been highlighted in chemically induced cell transformation [67,68,69,85], including TGF-beta, the master gene in the epithelial-mesenchymal transition (EMT). The EMT is involved in several biological processes in normal conditions including embryogenesis and wound healing in adults, but it also drives tissue fibrosis and cancer. In human cancer, the EMT is recognized as the committed step at tissue level that marks dysplasia progression and acquisition of invasive properties. EMT associated transcription factors (EMT-TFs) are different in physiological and pathological conditions. The recognition of EMT-TFs, such as Twist, Snail, Slug, and Zeb, is of prognostic value. At the cellular level, the EMT is characterized by changes in cell morphology from a round, cuboidal shape to spindle shaped cells, the disruption of cell-to-cell connections, the breakdown of the basal membrane, and the acquisition of migratory and invasive ability [88]. This entire process has been identified in in vitro cell transformation models. Transcriptomic-based studies highlighted the modulation of several EMT-TFs and other markers related to the transition to the mesenchymal phenotype [67,68,69,85]. The formation of aberrant colonies and malignant foci is characterized by a change in morphology (spindle-shaped cells). Finally, cells acquire chemotactic and invasive capacities [25,27,46,89]. Taken all together, these results show that the entire process of oncotransformation in vitro resembles the key events in human tumor progression and supports the use of CTAs as useful models to predict chemical carcinogenesis.

Interestingly, the sequence of events leading to EMT-linked progression to malignancy in the CTA does not appear to be limited to a specific class of chemicals, since the process is triggered by different chemicals belonging to various chemical classes. EMT-linked progression to malignancy resembles the late steps in the conceptualization of pathways leading to adverse outcomes, which by then are considered disengaged from the initial chemical exposure as the pathway progresses, while the initiating event and early key events are strictly related to the initial specific chemical insult. Indeed, transcriptomic-based studies, performed by using 3T3 cells, used three reference chemicals for the CTA, namely 3-methylcholanthrene (3MCA), benzo(a)pyrene (B(a)P) and 12-O-tetradecanoyl-phorbol-13-acetate (TPA), to induce cell transformation. For example, for B(a)P, it appears evident that the process is initiated by the activation of the aryl hydrocarbon receptor (AhR) and the modulation of its canonical pathway [67,68,69]. In another example of receptor associated MIE cell transformation activity, it has been observed that for SHE cells treated with diethyl-hexyl-phthalate (DEHP), the modulation of CYP2E1 may be related to the peroxisome proliferator activated receptor α (PPARα), which is recognized to be a target of DEHP. Landkocz et al. [85], describe how Cyp2E1 expression increased in DEHP treated cells (at a concentration range of 12.5–50 µM) versus untreated, but PPAR transcripts did not quantitatively differ between controls and treated cells, whatever the concentrations of DEHP and duration (5 h and 24 h) of exposure, for any of the PPAR isoforms.

The evidence that there is crosstalk between CYP2E1 and PPARα is supported by the fact that substrates of CYP2E1 may also serve as PPARα agonists. Indeed MEHP, the main DEHP metabolite acts as ligand of PPARα and PPARγ, but there is no concentration-dependent increase of PPAR expression upon DEHP treatment. The modulation of genes related to lipidogenesis also confirms that the model is able to highlight molecular events related to adverse effects triggered by DEHP [85]. In the SHE model, the specific targets of DEHP are an increased expression of NF-ΚB and Bcl2, concomitant with a decreased expression of p53 and c-myc [85,90], both of which are markers of antiapoptotic effects of DEHP in these primary cells. Therefore, the use of the CTA makes the identification of the early events specific to the tested chemical possible, from the main molecular target able to initiate the cell response, to the sustaining of the process, to the final outcome. The later/end key events triggered by these prior mechanisms are chemically agnostic.

### 2.4. Current In Vivo and In Vitro CTA Models Depict Different Steps of Carcinogenesis; How Can These Be Appropriately Integrated?

In this section we show how the CTA models can be used collectively to study multistep carcinogenesis: from the initiation/promotion protocols to transcriptomic-based models for identifying non-genotoxic carcinogens. Relevance, strengths and weaknesses in relation to human cancer hazard assessment are clarified.

#### 2.4.1. The Mouse Skin Model

The first experiments to understand the multistage carcinogenesis were performed in the 1940s using the mouse skin model [91]. Croton oil was applied in combination with polycyclic aromatic hydrocarbons (PAHs) to highlight the different steps leading to animal cancer. These experiments allowed the identification of a genotoxic initiating process, responsible for the conversion of normal cells into preneoplastic cells, whose replication when sustained by promotors such as croton oil, or in some cases complete carcinogens, leads to tumor formation. Most of the time initiators require only a single administration to cause a mutation whereas promoters and complete carcinogens (initiation and promoting properties) require prolonged administration to sustain the necessary cell proliferation to fix the mutations and achieve independent uncontrolled replication. Several aspects of the initiation/promotion theory were described: the specificity and irreversibility of the initiation change, and, conversely, the characteristics of non-specificity and reversibility of the promotion step.

Initial conclusions from these first reports are consistent with the current knowledge of the process leading to human tumor as a consequence of chemical exposure, including the concept of preneoplastic lesions that may or may not turn into neoplastic lesions, depending on the microenvironment.

Some important considerations with respect to these studies, however, should be noted. The combination of PAHs at low concentrations together with repeated application of croton oil (or similar irritating agents) result in the production of benign papilloma, in a reduced number of animals. Only a few of these papillomas progress to squamous cell carcinomas, and the rate of this transformation is low. Papillomas do not develop at all after insufficient application of the tumor promoter, or if the interval between individual applications is increased [92,93].

Subsequent experiments demonstrated that the tumor promoter TPA is often ineffective in the conversion of papillomas to carcinomas in the mouse skin model, while the treatment with other genotoxic chemicals is effective, suggesting that the progression to malignancy is related to the accumulation of mutations [94,95]. This also suggests that while the mouse skin model has provided new insights into the changes associated with initiation and promotion, used in isolation it is not a suitable model for studying initiation and promotion in carcinogenicity. To examine this further and elucidate whether there is a dose-response relationship in the tumorigenic effect of TPA, groups of hairless mice were treated topically on the back skin with different doses and treatment schedules of TPA in acetone. The study demonstrated that TPA alone induces a significant incidence of papillomas (*p* value 0.05) and some carcinomas in mouse skin. There is also a very significant dose-response relationship in the production of skin tumors in hairless ice after painting with various doses of TPA in acetone [95].

These observations seem to be more in agreement with the evolutionary paradigm of the Fearon–Vogelstein model for human carcinogenesis [63], showing that at least four steps are needed, with a minimum of five or six genetic changes to sustain the entire process arising from normal epithelium to the malignant tumor.

Cutaneous squamous cell carcinoma (cSSC) represents one of the most common cancers with an increased incidence. It originates from uncontrolled proliferation of atypical epidermal keratinocytes. A slow and gradual process from actinic keratosis to invasive cancer [96], it is rarely a metastatic cancer, and with treatment, it generally has a good prognosis. Ionizing and ultraviolet radiation are recognized as causal factors for cSSC, and age, male gender, light skin phenotype and immunosuppression are considered to be risk factors. At a molecular level, cSSC is characterized by an accumulation of mutations in several key genes, namely TP53, EGFR, RAS, with certain mutations driving the progression from premalignant to malignant forms [97]. At the cell and tissue level, cSSC is characterized by cell heterogenicity with cells at a different stage of differentiation and different rate of proliferation, which can even be reversed to a quiescent state. Contrary to other epithelial tumors, squamous differentiation in cSSC represents a protective role by increasing tissue resilience. However, if metaplasia persists, the risk of dysplasia and cancer increases [98]. The mouse skin model seems to resemble this process.

All the key papers published in this field correctly address the model as a tool to study the multistage process of cancer and identify the steps of initiation, promotion and progression [93].

#### 2.4.2. Revisiting the Concept of Promotion

The term ‘promoter’ has been used in relation to the properties of a chemical to sustain the formation of papilloma and the progression to carcinoma, independently of its mode or mechanism of action.

The croton oil or TPA were chosen as ‘promoters’ for their effects in skin diseases and disorders. Indeed, croton oil is an irritating chemical, inducing an acute inflammatory response characterized by vasodilatation, polymorphonuclear leukocyte infiltration to the tissue and oedema formation. PAHs can also trigger an immune response characterized by the production of inflammatory chemical and cellular mediators. It is possible that this combination of effects is responsible for the formation of the preneoplastic lesions. At concentrations that exceed the ability of the detoxification mechanisms, PAHs electrophilic metabolites can form DNA adducts and act as complete carcinogens. However, in the pioneering initiation/promotion studies, PAHs were administered at concentrations that did not induce neoplastic lesions after a single application [91,99].

TPA belongs to the chemical class of tetracyclic diterpenoids known as phorbol esters. As with all the members of this class, TPA mimics the action of diacylglycerol (DAG). DAG is a second signaling mediator produced downstream of the activation of several receptors, including tyrosine kinase receptors. DAG is produced by members of the phospholipase C family that cleave phosphatidyl-inositol(4,5)bisphosphate to inositol triphosphate and DAG. DAG is then transformed in phosphatidic acid by diacylglycerol kinases (DGK) in T cells [100,101,102]. The role of this process in cancer is well known, and more recently it has been associated with immune evasion mechanisms and with the inactivation of RAS-mediated signaling [103]. Therefore, TPA dysregulates the signaling cascade following DAG activation, by acting on the same pathways. A recent report, showing a key role of DAG-mediated response to several stress conditions in plants, reveals an evolutionary interplay between tetracyclic diterpenoids and DAG [104], reinforcing the hypothesis of the disrupting effect of TPA on DAG-mediated pathways in animal cells. However, this is a unique mechanism related to the chemical structure of the phorbol esters.

#### 2.4.3. The Rat Hepatocarcinogenesis Model

The rat hepatocarcinogenesis initiation/promotion model was first proposed by Peraino and co-authors in 1971 [105]. Several models have been developed since then [106,107,108,109]. All of them include the initiation with an initiating carcinogen, followed by the treatment with a promoting substance, and a second treatment with a progressor. A partial hepatectomy is often applied. An example of the rat hepatocarcinogenesis model included the treatment of rats starting at 5 days of age with an initiating agent, usually diethylnitrosamine or ethylnitrosamine, a treatment with a promoting agent, usually phenobarbital, at weaning, a partial hepatectomy at 6 months, and then the treatment with putative progressor agents, usually another genotoxic chemical, such as ethylnitrosourea [110]. The treatment scheme sustained the formation of altered foci in hepatocytes. Hepatocarcinoma developed only if the promoting substance was continuously administered. The model was proposed as a replacement of the rodent bioassay to identify chemicals acting as promoters. Several chemicals have been tested in this model, including hormones and hormone-like substances [111].

The formation of foci of altered hepatocytes is considered a relevant endpoint of the hepatocarcinogenic effect, but it has never been considered relevant to humans [112].

#### 2.4.4. Two-Stage Cell Transformation Assay

The two-stage, initiation/promotion cell transformation assay was developed in the 1950′s to resemble the mouse skin model. The reference schedule includes an initiating treatment with a carcinogen, usually 3-MCA or B(a)P, at low concentrations, followed by the treatment with TPA as a promoter.

A similar initiation/promotion scheme was applied in pioneering in vitro studies using embryo rodent cells [113,114].

Amongst the CTA models currently in use, the BALB/c 3T3 model was the first transformation assay to be set up to examine the tumor promotion in vitro [24]. A Level-II amplification protocol was also proposed for this model. In the amplification protocol, cells treated with the tested chemical are passaged when they reach the confluence required to prevent cells from becoming quiescent, allowing further replicative cycles [115,116]. This resembles the partial hepatectomy in the rat hepatocarcinogenesis model. The Balb c/3T3 CTA has a history of two step models that were developed before the one step model. A first highly sensitive model was investigated by modifying the medium [117], whilst another with a partially modified protocol [118] has been reported, and interlaboratory reproducibility has been verified [119,120].

The paradigmatic example for the use of the two-stage initiation/promotion CTA to highlight non-genotoxic carcinogenic chemicals is represented by the study on okadaic acid (OA) and dinophysistoxin-1 [121]. In this study, cells were treated with a low concentration of 3-MCA, as the initiating agents, and then with OA or dinophysistoxin-1. Both chemicals were able to enhance the cell transformation in 3-MCA-treated cells. OA was also tested in two-stage carcinogenesis experiments involving mouse skin, by dermal administration, and mucosa of the rat glandular stomach by oral administration, showing very potent tumor promoting activity. While the mechanisms of acute toxicity exerted by OA and its analogs, the dinophysis toxins, are well known, much less is known about the mechanism sustaining its promoting activity. OA and its analogs are diarrheic shellfish poisoning (DSP) molecules, and besides DSP syndrome, they can exert neurotoxicity, and recently have also been associated with Alzheimer’s disease [122,123].

OA has been associated with protein phosphorylation: it is a potent and selective inhibitor of protein phosphatase, PP1 and PP2A. The molecular interplay of PP2A and MAPK pathways has been suggested as a key event in OA-mediated neurotoxicity and possibly in OA-mediated promoting effects. Even if OA was unable to induce cell transformation in the absence of 3-MCA treatment, several reports show that OA can induce genotoxicity via micronuclei formation, oxidative DNA damage, sister chromatid exchanges, 8-hydroxy-deoxyguanine adducts, minisatellite mutations and DNA strand breaks [122]. It has also been claimed to be an aneugenic agent. Results are often contradictory and, interestingly, they are related to cell type and experimental conditions.

The recent publications reporting upon the molecular events occurring in the BALB/c 3T3 CTA, at different timepoints and concentrations, provide food for thought. To recap, in the initiation/promotion scheme, carcinogens, such as 3-MCA, are used at low concentrations to ‘initiate’ cells, without inducing fully malignant transformation. In fact, 3-MCA at low concentrations is successfully detoxified, but oncotransformation is sustained only when critical higher concentrations of carcinogens are applied. Importantly non-genotoxic key events here precede the genotoxic event, and 3-MCA is a chemical classically understood to have a genotoxic mechanism [67]. It is only with early mechanistic information that it becomes possible to fully elucidate the type of interplay between ‘initiation’ and ‘promotion’, at the molecular level, and the contribution of a specific chemical to sustain the process of cell transformation.

This limitation is overcome in the Bhas 42 CTA. Bhas 42 cells are initiated by multiple copies of activated RAS. The role of RAS in the cell signal transduction is well described at the molecular level. The activation of the oncogene RAS in human tumors marks the step to malignancy, but other key events are required for tumor progression. Most of these events are orchestrated by the microenvironment and can be fostered by ‘promoters’. However, chemicals that ‘promote’ the neoplastic transformation can be either genotoxic, since tumor progression is associated with the accumulation of mutations, or non-genotoxic, through the disruption of molecular pathways regulating cell and tissue homeostasis, or, as with 3-MCA, they can operate via both non genotoxic and genotoxic mechanisms.

The Bhas 42 CTA is a good model to highlight the ability of a chemical to sustain the progression to malignancy. It better resembles the late steps that mark the progression from preneoplastic to neoplastic lesions in human. For the Bhas 42 CTA, the ‘promotion’ protocol is more relevant for human carcinogenesis, as the cells are already initiated.

The recent study by Ohmori et al. [69] confirms that Bhas-42 can be used to identify non-genotoxic carcinogenic chemicals, such as TPA, without applying an initiating/promotion protocol. Whilst there is little evidence that TPA can act as a complete carcinogen in a normal epithelium in the mouse skin multistage carcinogenesis model (see discussion in Section 2.4.1), not surprisingly, it can induce oncotransformation in Bhas 42 cells. This is via a mechanism involving AhR as an initiating event and the dysregulation of cancer-related genes as early key events [69], thereby confirming the Bhas 42 CTA as a suitable model for the study of the later steps in cancer progression. In the Bhas 42 CTA, many NGTxCs other than TPA are judged to be positive [124].

Other studies, however, show the possibility of using SHE or BALB/c 3T3 models to study chemicals recognized to be carcinogens via non-genotoxic mechanisms, without applying an initiation/promotion schedule. An overview of these studies, as discussed and evaluated using stringent evaluation criteria agreed upon by the OECD CTA expert group, is reported below.

#### 2.4.5. Overview of Chemicals Tested in SHE and BALB/c 3T3 Models Recognized to Be Carcinogens versus Non-Genotoxic Carcinogenic Mechanisms

In 2014, the OECD CTA expert group agreed on the need to refine the categorization of genotoxic versus non-genotoxic chemicals which have been previously tested in the CTA [37]. This categorization is published here for the first time. In order to discriminate between genotoxic and non-genotoxic carcinogenic chemicals, the OECD CTA expert group agreed on adopting the EU Reference Laboratory European Centre for the Validation of Alternative Methods (EURL ECVAM) criteria as previously described to characterize genotoxic and non-genotoxic chemicals [125,126]. The list of in vitro and vivo genotoxicity tests, together with the curated categorization considered in the evaluation process, is reported in Table 1 and Table 2. After the initial evaluation by the OECD CTA expert group, the data were also updated considering the more recently published and curated EURL ECVAM Genotoxicity and Carcinogenicity Consolidated Database of Ames Negative Chemicals [127]. It is notable that with these recent updates, several chemicals previously considered to be negative for genotoxicity (but positive for carcinogenicity) are now reported to be positive for some genotoxicity tests (Table 3, Appendix A).

Based on the criteria reported in Table 1, a list of non-genotoxic chemicals which have been tested in the SHE and/or BALB/c 3T3 CTA models and reported in OECD 2007 [37] was derived. For these chemicals a complete evaluation of the carcinogenic and transforming properties was performed, based on the information obtained from several data sources (Table 2).

This then became the basis of the evaluations provided in Table 3, Table 4, Table 5, Table 6 and Table 7, supplemented with more recent epidemiological and mechanistically relevant references. The classifications of A, B, C, and D are based upon the evidence of carcinogenicity, as shown in Table 6.

#### 2.4.6. Ability of the SHE CTA to Identify Carcinogens with Non-Genotoxic Properties

On the basis of the OECD CTA expert group review as presented in Section 2.4.5, and Table 2, Table 3 and Table 4 and summarized in Table 6, it is possible to identify a list of non-genotoxic carcinogenic chemicals that induce SHE cell transformation.

Of a total of 16 chemicals, thirteen chemicals were tested at pH 6.7. Three of these chemicals, DEHP, TPA and clofibrate were also tested at pH 7.0. Three other chemicals, benzoyl peroxide (BPO), methylclofenapate and caprolactam were tested at a pH 7.0 (Table 3). Caprolactam tested negative at pH 6.7 [44]. Shortly after, IARC concluded that for the data for caprolactam results on morphological transformation on mammalian cells were inconclusive [181].

The list includes methyl eugenol, a flavoring agent, whose mode of action as a genotoxic or non-genotoxic carcinogenic chemical is still debated.

Methyl eugenol has been tested in several genotoxicity tests [182]. Its metabolites are reported to be able to induce protein and DNA adducts. It can also induce unscheduled DNA synthesis. Recently methyl eugenol tested positive in in the in transgenic rodent somatic mutation assay and in the in vivo comet assay [127] and gpt-Delta transgenic rats following medium-term exposure [182]. However, results from in vitro genotoxicity tests, supporting genotoxicity classification, including the Ames test, chromosomal aberrations in CHO cells and micronucleus test, were negative [182]. The carcinogenic properties of methyl-eugenol observed in rodents are strictly related to the treatment dose, with a threshold at 37 mg/kg bw. This result and the well-known ability of methyl-eugenol to induce oxidative stress via the formation of free radicals, are suggestive of a non-genotoxic mechanism of carcinogenesis, which leads to tumor formation only when the pro-oxidant-antioxidant balance in the cell is disrupted. For this reason and considering the negative results in in vitro genotoxicity tests, methyl-eugenol has been included in the list of non-genotoxic carcinogens.

On the basis of the available information, chemicals were clustered into four correlation groups (A to D), according to the level of confidence between the transforming properties in the SHE CTA and the evidence of carcinogenicity in animal studies (Table 3 and Table 4).

Group A includes the chemicals for which a clear correlation with carcinogenicity exists. However, for all chemicals in this group, the evidence of carcinogenicity has been long debated. Results derived from the RCB are often equivocal or inconclusive, based on treatment schemes, number of animals, and durations of treatment that are not always aligned with OECD test guideline specifications. Four of the chemicals in Group A, DEHP, methyl eugenol, oxymetholone and reserpine have been classified as ‘reasonably anticipated to be human carcinogen’ in the 14th and 15th Report on Carcinogens by the US National Toxicology Program [129]. The criteria to determine whether a chemical can be included in this category considers the availability of human and animal data, i.e., limited evidence in human studies or sufficient evidence from animal studies; or limited evidence in human and animal studies for chemicals or mixtures belonging to structurally related chemical class, whose members have been already classified as known or possible carcinogens to humans; or there is convincing information that the mechanism of carcinogenesis is human relevant. The same criteria have been adopted to include DEA, DETU and ethylbenzene in Group A.

With the exception of oxymetholone and reserpine, for which limited evidence of cancer in human has been reported, all the other chemicals induce cancer in animals through mechanisms and mode of action where human relevance is debated.

Oxymetholone is an anabolic androgenic steroid (AAS) that was mainly marketed to treat certain types of anemia caused by deficient red cell production [183]. As all AAS, oxymetholone can increase muscle mass, a property that has propelled its use in the treatment of skeletal muscle wasting in chronic diseases, including cachexia in HIV/AIDS patients [184]. The extensive use, misuse and abuse of AAS in athletes to improve physical performance, or for aesthetical purposes, has enriched the scientific knowledge of the detrimental effects that prolonged and high dosage treatment with this class of drugs may exert on many organs and physiological functions [145]. AASs have been recognized as carcinogens acting via different mechanisms, including the modulation of the androgen receptor or the estrogen receptor [145]. Oxymetholone is a member of the subclass of 17α-alkylated anabolic-androgenic steroids. It is a steroid with a relative low affinity with the androgen receptor but with high myotrophic-androgenic index, which confers a high anabolic to androgenic ratio [185]. Oxymetholone has been reported to exert estrogenic activity, inducing gynecomastia and water retention, through the direct activation of the estrogen receptor [183,185]. Therefore, it may be assumed that oxymetholone acts as an endocrine disruptor.

Hepatotoxicity is one of the main adverse effects elicited by AAS, of which cholestatic jaundice provides the major hepatic side effect [183,186]. These adverse effects are essentially related to orally administered AAS, such as oxymetholone [185]. Cholestatic jaundice in adults is mainly represented by primary biliary cirrhosis and sclerosing cholangitis, two inflammatory conditions that may evolve to neoplastic lesions, periampullary carcinoma and cholangiocarcinoma [187]. Cases of cholestatic jaundice have been reported in patients treated with oxymetholone, including a case of periampullary carcinoma, as well as hepatic tumors and esophagus tumors [183,188]. Benign and malignant liver tumors have been associated with the use of AAS, strictly related to dose and duration of the treatment [183]. Neoplastic growth can be reverted by the suspension of the treatment, an occurrence that is typical of tumors induced by non-genotoxic carcinogens. Liver is the main target in animal carcinogenesis. Oxymetholone has been orally administered to F344 rats, both sexes, a statistically significant increase of hepatocellular adenoma, carcinoma and hepatoblastoma was observed only in females at the highest tested dose (71 mg/kg bw/day) [129,189]. The fact that oxymetholone was tested only in rats, and not in mice, drastically reduced the possibility to give evidence for cholestatic jaundice–related lesions, since rats lack both gallbladder and bile ducts [190].

Reserpine is an indole alkaloid extracted from the roots of Rauwolfia serpentine, it is used in clinical practice as an anti-hypertensive drug and tranquilizer. The hypotensive and sedative effects are related to the ability of reserpine to block the uptake and storage of serotonin, norepinephrine, and dopamine into presynaptic storage vesicles, leading to their depletion from peripheral and central synapses [191]. Reserpine has been associated with an increase of breast cancer incidence in long-term treated patients [146]. Mammary glands are the targets for experimental carcinogenesis in the female mouse [129,192]. Reserpine also induces seminal vesicle undifferentiated carcinoma in the male mouse and adrenal pheochromocytoma in the male rat [129,192]. Reduction or inhibition of dopamine has been related to the increase of prolactin. Prolactin has already been reported to play a key role in triazine rat mammary carcinogenesis [193]. The contribution of prolactin to the pathogenesis and progression of human breast cancer has also been extensively debated [149,194], especially in post-menopausal women [195]. It is possible to speculate that the antidopaminergic activity of reserpine is responsible for the increase of the level of circulating prolactin. In rodents, hyperprolactinemia is related to mammary gland hyperplasia, which evolves to dysplasia and cancer. To the contrary however, for normal cells, mammary gland hyperplasia is a benign breast condition in humans. Hyperprolactinemia in humans affects the prolactin receptor-mediated signaling pathways that are shared with members of the cytokine receptors superfamily [196]. Indeed, hyperprolactinemia is also a marker of disease progression and poor prognosis. [149,196]. Therefore, it seems that reserpine-mediated hyperprolactinemia is a key event (or perhaps an initiating event) in both rodent and human breast cancer. It remains to be discovered whether the mechanism of cell transformation by reserpine in SHE cells is due to the activation of signals related to the modulation of the prolactin receptor.

DEHP, the prototypical member of the chemical class of phthalates, induces hepatocarcinoma in rat and mouse, through the activation of the PPARα, a mechanism for which human relevance is highly debated. Other chemicals included in the list of non-genotoxic carcinogens, such as Wyeth-14,643, BBP and clofibrate, share the same mechanism of carcinogenesis in rodents (Table 4). While the role of PPARα in fatty acid metabolism is well recognized in both human and rodents, its role in cancer shows high species-specificity. Indeed, only non-malignant adverse effects were reported in patients treated with clofibrate, a drug used to treat dyslipidemia in cardiovascular disease. Some literature reports even suggest the use of PPARα activators, such as Wyeth-14,643, another hypolipidemic chemical and potent peroxisome proliferator, to prevent and treat certain types of human cancer [197]. DEHP, BBP, clofibrate and Wyeth-14,643 were tested for carcinogenicity in several models of transgenic animals. Clofibrate tested negative for neoplastic growth in p53+/− heterozygous mice and in Tg.AC mice. After oral administration [155,198], it induced a slight increase in the incidence of several neoplasms in rasH2 transgenic female mice [199], and following dermal application tested positive in Tg.AC mice, inducing a dose-dependent increase in the incidence of cutaneous papillomas [200]. DEHP and Wyeth-14,643 were tested in PPARα-null mice to determine whether their carcinogenic properties were dependent on peroxisome proliferation [131,201]. In PPAR α wild-type mice, Wyeth-14,643 was found to induce multiple hepatocellular adenomas and, sometimes, carcinomas, while it tested negative in PPARα-null mice. DEHP is considered a less potent carcinogen than Wyeth-14,643. However, in contrast to Wyeth-14,643, DEHP can induce hepatocarcinogenesis in PPARα-null mice, suggesting that a different mechanism may be involved in DEHP carcinogenesis [201]. The possible involvement of the AhR and of its canonical pathway has been suggested [132], as well as endocrine and epigenetic mediated effects (as reviewed in [74,173]).

Even if PPAR receptors, specifically PPARα and PPARγ, are highly expressed in human and rodent embryonic cells, the process of oncotransformation of SHE cells by DEHP seems to proceed independently of PPAR activation [85]. This process has been reported to be related, at the molecular level, to the modulation of genes and pathways involved in cytoskeleton remodeling, a key event in SHE CTA [20,85]. Within 24 h hours of SHE exposure at a concentration starting at 25 µM, DEHP perturbed cytoskeleton regulation and up-regulated genes related to transcription factors and effectors involved in cell proliferation, apoptosis and transformation (*PI3kinase*, *homeobox1*, *HOXA10*, *TNF-α*, *NF-ΚB* and *TGFβ+*), while it down-regulated *c-myc* and *p53*, as well as interactions with the thyroid hormone receptor and methyl transfer reaction ((S-adenosyl homocysteine hydrolase) [85]. Up-regulation of the genes *Tnf-α*, *Nf-κB* as well as *Pla2g2d* (Phospholipase A2 group IID), known to be implicated in pro-inflammatory mediation [202,203], indicates an early immune response that may be related to SHE cell transformation and tumor promotion [204,205].

As far as the ability of caprolactam to induce cell transformation in the SHE model is concerned, this chemical has not been recognized as a possible carcinogen. It was first included in IARC classification group 4, as possibly not carcinogenic to humans, but was recently included in group 3, following the recent update of IARC evaluation procedures for identifying cancer hazards [206]. On the basis of divergent results, at different experimental conditions [44,207], IARC [181] stated that the ‘results for morphological transformation in mammalian cells were inconclusive’. In the absence of mechanistic information, which could explain caprolactam transforming properties in SHE CTA, this result should therefore be considered as a false positive.

#### 2.4.7. The BALB/c 3T3 CTA Performance in Identifying Non-Genotoxic Carcinogenic Chemicals

Using the same approach as described above, here we evaluate the ability of BALB/c 3T3 CTA to induce cell transformation via non-genotoxic carcinogenic mechanisms as performed in the SHE CTA.

Results are reported in Table 5, Table 6 and Table 7. The initial list of chemicals was obtained from OECD DRP No 31 [37]. Non-genotoxic chemicals (both carcinogenic and non-carcinogenic) were selected on the basis of the GHS criteria and the criteria shown in Table 2. Among all 10 chemicals that have been reported to induce cell transformation in the BALB/c 3T3 CTA via non-genotoxic carcinogenic mechanisms, we reduced the list to 6 chemicals, for which mechanisms of action are discussed in Table 7. Two chemicals, ethanol (ethylic alcohol) and sodium saccharin, were excluded for the following reasons:

Ethanol has been classified as carcinogenic to humans on the basis of a plausible but not fully supported analysis from an epidemiological study regarding the chronic assumption of alcohol and cancer at different organs in alcoholics [208]. However, it is still debated as to whether cancer initiation is related to the genotoxic properties of acetaldehyde, the main metabolite of ethanol or, as is more likely, by immune-mediated inflammation, through non-genotoxic events [208]. Sodium saccharin has been declassified as a carcinogen by many (but not all) scientific and regulatory agencies, including IARC and NTP.

Additionally, mechanisms of methylcarbamate, cinnamyl anthranilate and mezerein are not discussed here, since the data are inadequate to draw conclusions on carcinogenic properties and mechanisms of action.

The remaining substances were grouped according to the level of evidence for carcinogenicity, already described. Two chemicals from the short list in Group A, DEHP and reserpine, have already been discussed in Section 2.2, as they have also been tested in the SHE CTA.

The rationale for the inclusion in Group A for the other chemicals is briefly discussed below.

Diethyl stilbestrol (DES) is a well-recognized human carcinogen [129]. It has been used in the past to prevent miscarriages, treat prostate cancer and stimulate cattle growth. It is not widely known that DES was initially used for treating estrogen deficiencies, such as vaginitis and symptoms of menopause, and for postpartum lactation suppression. Ever since it became available on the drug market, this synthetic estrogen was postulated to induce cancer. However, it was only due to extensive use to prevent miscarriages over a 50-year period that the medical and scientific communities became aware that DES is one of the most potent carcinogens, disrupting the epigenetic programming of offspring, eliciting transgenerational adverse effects (reviewed in many publications including Jacobs et al. [173]). Indeed, DES induced clear-cell adenocarcinoma of the vagina and cervix in young women (DES-daughters), whose mothers took DES during pregnancy, but it is also thought to increase the risk of breast cancer in second (DES-grand-daughters) and third (DES-great grand-daughters) generations. As a synthetic estrogen, DES can react with the estrogen receptors, but the molecular mechanisms triggering and sustaining the pathway to cancer are still poorly understood.

Dichlorodiphenyl trichloroethane (DDT) has been widely used in the world as an insecticide and it is still used in tropic and subtropic regions to control insect borne diseases such as malaria. DDT and its metabolites are highly persistent in the environmental media and in biological fluids of directly and indirectly exposed populations. Despite its widespread use for many years, its bioaccumulation and persistence, the epidemiological evidence for DDT-related adverse outcomes is not so clear [209]. However, DDT is considered a possible human hepatocarcinogen, based on results in animal studies [129,210]. Its mechanism is related to the activation of constitutive androstane receptor (CAR), sustaining the induction of hepatic microsomal enzymes [210], a mechanism whose relevance to humans is still debated [4,211]. However, gene signatures detected in hepato-proliferative lesions in male F-344 rats exposed to DDT are consistent with the dysregulation of cell cycle and the induction of mitogenic signals [210].

Estrogen is classified as a carcinogen for its use in menopause therapy. Conjugated estrogens are most commonly used for hormone replacement in women who have entered menopause primarily due to hysterectomy. Epidemiological studies gave evidence for a significant increase in the incidence of endometrial and ovarian cancer as a consequence of the therapy. Estrogen metabolism is substantially mediated by Cyp1A1 and Cyp1B1, showing an important molecular interplay with AhR-mediated canonical signaling. BALB/c 3T3 cells express both AhR and ER, showing high sensitivity to the transforming properties of estrogens [10,67,179].

Nitrilotriacetic acid, trisodium salt (NTA) is a metal ion chelating agent primarily used in laundry detergents. Therefore, the general population may be exposed through the ingestion of drinking water or through dermal contact. However, there is no epidemiological evidence of a possible relationship between human cancer and exposure to NTA. Benign and malignant renal tumors have been found in animals exposed to NTA, including adenoma and adenocarcinoma of the kidney and transitional-cell carcinoma of the kidney, ureter and urinary bladder [129]. The mechanism(s) by which NTA induces tumors has not been adequately explored. It has been suggested that the nephro-carcinogenic properties of NTA are related to dose-dependent changes in intracellular zinc ions homeostasis, due to the chelating properties of NTA [129].

#### 2.4.8. Bhas 42 CTA Performance in Identifying Non-Genotoxic Carcinogenic Chemicals and Mechanistic Studies

Table 8 shows 22 chemicals with positive results in the stationary-phase test of the Bhas 42 CTA that were negative or not definitely positive in the Ames test, and other genotoxicity tests [39,124,212,213].

Sixteen of the chemicals in Table 8 have been classified as Class 1 or 2 by IARC, as carcinogens by NTP, or as tumor-forming chemicals. Among them, methapyrilene HCl, which is a hepatocarcinogen in rats, is also included. In addition, in a two-stage carcinogenicity test in animals, chenodeoxicholic acid, cholic acid, deoxycholic acid and lithocholic acid are reported to be promoters of colon cancer, and mezerein, OA, and TPA are reported to be promoters of skin cancer. Application of TPA alone has been suggested to be a skin carcinogen.

In the Bhas 42 CTA, the mechanisms of 10 chemicals out of a total of 22 have been reported as follows:

For the nine chemicals classified as NGTxC, DNA methylation in the cells of the formed foci has been analyzed. In cells’ transformed foci by cadmium chloride and lithocholic acid, pathway enrichment analysis revealed that genes harboring hypermethylated differentially methylated regions (DMRs) were significantly enriched pathways including pathways in cancer, basal cell carcinoma and Wnt signaling [212].

Additionally, DNA methylation in cells transformed foci by each of cholic acid, diethanolamine, DEHP, methapyrilene hydrochloride, OA, sodium saccharin and 2,3,7,8-tetrachlorodibenzo-p-dioxin have been analyzed. In the pathway analysis, the genes with DMRs at the CpG sites were found to be enriched in cancer-related categories, including ‘cell-to-cell signaling and interaction’ as well as ‘cell death and survival’. Moreover, the networks related to ‘cell death and survival’, which were considered to be associated with carcinogenesis, were identified in 6 NGTxC chemicals [213].

In the Bhas 42 CTA transcriptomics analysis conducted over four time points, many genes (specifically 2289) with statistically significant changes in the expression were detected during the process of focus formation by TPA, of which 2648 genes were downregulated. Further analysis of the genes whose expression altered during the process of focus formation by TPA, revealed that not only RAS-related genes and signals were involved, but also the expression of many ‘Hallmark of Cancer’ genes and signals were altered [69].

In addition to TPA, Ohmori et al. also obtained transcriptomics data collected at multiple points over time in three complete replicate experiments for more than 10 NGTxC, several GTxC and other Bhas 42 CTA-positive chemicals. For many of these, proteomics data from protein samples were also obtained and data are currently undergoing analyses. The Bhas 42 CTA has been approved and declassified as a guidance document by the OECD [54] having demonstrated that it is highly robust and reproducible.

## 3. Discussion and Next Steps

The results reported in Table 1, Table 2, Table 3, Table 4, Table 5, Table 6, Table 7 and Table 8 illustrate that the CTA(s) can detect non-genotoxic carcinogens. Whilst several in vitro tests have been developed and validated to classify genotoxic chemicals, in vitro data are not considered in the current criteria for the classification of carcinogenicity. Therefore, even if the CTA represents a unique in vitro test that can provide an endpoint of morphological transformation, it is not considered sufficient on its own to be adequate to classify chemicals as carcinogens. However, it is a particularly useful component of the IATA for NGTxC. This has inspired and prompted several mechanistic studies to open the ‘black box’ of key events linking chemical exposure to cell malignancy, with the intention of increasing regulatory confidence in the CTA, as a key component of a tiered and integrated testing strategy for detecting NGTxCs.

As described before, in developing an IATA for NGTxCs, it is crucial to consider all relevant endpoints of adversity [4].

The two models using 3T3 cells show that in vitro oncotransformation is sustained by similar gene pathways, marking key processes related to immune-mediated responses [67,69]. However, the same key signaling pathways become active at different times in the two models, confirming that Bhas 42 cells are at a later stage of the transformation process.

The currently available studies do not allow a complete comparison of the molecular events in the two 3T3 CTA models, due to differences in the mechanisms of the tested chemicals and experimental time schedules.

The overall picture, however, gives evidence for some key biological processes that can be detected in the CTA models, including the activation of the metabolic pathways, the key pathways in innate immune response, the turning point related to the saturation of adaptation mechanisms, and, primarily, the molecular events that are strictly related to critical changes at the tissue level, such as the process of EMT. The early steps of this process are marked by the cell cytoskeleton alteration, particularly evident in the primary SHE cells, after 24 h-chemical exposure. Critical molecular signaling in cell proliferation, leading to the critical changes in cell morphology and sustaining in vitro oncotransformation in all three CTA models, aligned with the transcriptomic prescreening approaches [74,214] are reported in Figure 5. The selection of the relevant CTA for the testing paradigm will be indicated on the basis of the previous MIE and prescreening tests conducted.

On the basis of the synthesis and critical review of the primary biomarkers identified and from relevant project work [67,68,69,74,214,215], the three CTA models are exemplified in the manner in which they can individually address the mechanisms and hallmarks of NGTxC as specified in the OECD IATA for NGTxC [3]. For the MIEs, the SHE CTA is able to address the cellular metabolism and receptor mechanisms of AhR signaling via CYP1B1, CYP2E1, epoxide hydrolase 1, GSH transferase and thioredoxin reductase. The BALB/c 3T3 has additionally been demonstrated to identify Ugt 1a, and Bhas 42 can also identify the AhR signaling via CYP 1A1 and 1B1. With respect to the pivotal KEs, of immune mediated inflammation, mitogenic signaling, cell injury and cell senescence, the SHE and Bhas 42 CTA KE of mitogenic signaling can identify a majority of different pathways, for the SHE: MAPK3, MAPK4, MAPK5, Ras oncogene family members and Homeobox 1, as well as NF-kB, and the Bhas 42 can identify several (IL-1, IL-2, IL-6 and TNFR2), but the BALB/c 3T3 can identify many (including classical and alternative complement pathway, IFN, IL-1, IL-4, IL-6, IL-9, IL-17, IL-18 and TNFR2). All the CTA models can identify various cell injury mechanisms, but only the SHE CTA and the Bhas-42 have been shown to pinpoint senescence bypass and telomerase signaling. With respect to sustained proliferation, again a variety of markers are clearly identified, the majority for the SHE and BALB/c 3T3, all differing between these two models. For the Bhas 42 model, intergin signaling and JAK/STAT signaling has been reportedly identified. All the indicated markers for sustained proliferation alter the tumor micro-environment and also the cell adhesion and cytoskeleton, which together can lead to oncotransformation.

Although it is still difficult to draw any definitive conclusions on the ability of CTA to feature the molecular initiating event(s), the involvement of the receptor-mediated activation of the metabolic pathways sustaining mechanisms of bioactivation and detoxification of xenobiotics is evident. Therefore, it is valuable to better characterize CTA models for the expression of receptors recognized as the main targets of specific chemicals which play a role in chemical mode and mechanism(s) of action. In parallel time, other in vitro tests targeting specific endpoints, such as the profile of inflammation-related interleukins and chemokines activated by the chemical exposure and/or the level and efficiency of cell-to-cell communications, may be considered.

Going forward it is important to reconsider the results of past CTA experimental studies; whilst they may have been unclear at that time, they can now be highlighted, on the basis of our improved mechanistic understanding of functional changes gained over the last decade. The application of transcriptomics tools and insightful experimental design have enabled and continue to enable this transition. We can start to recommend specific key event biomarkers (Figure 5), and now also specific differentially methylated regions, as preserved sustained proliferation epigenetic markers for senescence bypass in the SHE CTA [216].

The integration with other tests would be useful to confirm and refine the results obtained at the molecular level in the CTA, leveraging the CTA morphological endpoint of oncotransformation as the phenotypic anchoring of proxy biomarkers. Next, reproducibility of these results needs to be established by independent laboratories, to ensure confidence for regulatory applications.

In the intervening time since the OECD IATA NGTxC expert group was established in 2016, to date, several CTA method elucidation goals have been achieved to refine and facilitate a more consistent approach, for testing putative non-genotoxic carcinogens. However, some improvements that have been attempted are still far from completion. One of these is exploratory work to replace fetal bovine serum (FBS).

Serum is one of the most important components of cell culture media and acts as a source of basic nutrients including albumins, growth factors and growth inhibitors. Cell culture media are usually added with a 10% concentration of FBS to promote the growth of cell cultures. Lower concentrations, usually 5 or even 2%, promote the growth of transformed cells, since tumor cells become progressively independent of certain cell nutrients. Therefore, serum becomes a critical issue for the CTA. Both SHE and 3T3 cells are prone to be transformed if they are maintained in culture after reaching the confluence. Cell growth, however, depends on the quality and concentration of serum. Serum restriction may promote the growth of transformed cells in chemical-treated cultures. By contrast, a serum richer in nutrients may accelerate the proliferation in untreated cells leading to the selection of transformed clones. Therefore, it is essential to test each serum batch before use as there can be a lack of uniformity in the composition of the serum, and this variability can affect the final experimental outcome. In spite of this, it must be acknowledged that, from a general point of view, using FBS offers more advantages than disadvantages. FBS at 10% concentration is recommended in CTA-validated protocols.

However, from animal welfare perspectives, the use of FBS is criticized, as is the use of human serum [217]. Moreover, the recognized need to reduce the carbon footprint of cattle farms means that bovine serum provisioning may become more limited in the future.

Consequently, in recent years, alternatives to FBS supplementation in cell cultures have been considered, including synthetic serum and human serum or human platelet lysates. This is starting to be explored in relation to the CTA.

Table 9 provides experimental results for the culture of BALB/3T3 cells, clone A31-1-1 using commercial defined media, as compared to FBS, as part of a much larger study aiming at finding suitable alternatives to FBS. The results refer to the effects of one of a few commercially available synthetic serums on BALB/c 3T3 cell growth. From this study, it is evident that cell ability to grow in a culture medium strongly depends on nutrients of animal origin, as compared to the defined media tested.

Human serum has been considered a possible alternative to FBS. Human serum is successfully employed for medical cell-based therapy. Several critical aspects of the use of human serum for OECD test method and Test Guideline purposes, including ethical aspects, serum uniformity from different donors, transmission of known and unknown diseases, competition with medical uses and approaches as to how to address these, are being discussed [216]. Chemical contamination of human serum is also a potential confounder when used in in vitro chemical hazard test methods.

At present, the options to replace FBS for the culture of cells used in the CTAs need a great deal more work.

## 4. Conclusions

Almost 60 years after the first cell transformation model for testing chemical carcinogenesis was reported in the scientific literature, the CTA still appears to be the only in vitro model that provides an endpoint of oncotransformation. The advent of new omics technologies has enabled the opening of the ‘black box’ with the extraction of critical mechanistic information, to highlight the multistep process leading to in vitro oncotransformation. Taken together, this new information gives evidence for key events that are coherent and consistent with those described in the multistep carcinogenesis process in humans.

Human cancer onset is a long and complex process that requires overcoming several biological mechanisms of defense, repair, resetting, recovery, and re-establishment of homeostasis. The CTA can capture several of those key events that lead to the disruption of biological characteristics, known as cancer hallmarks, and to the committed step that marks the point of no-return leading to malignant transformation. It is this point of transformational no-return that is the critical step for human health protection against chemical induced carcinogenesis. Earlier key events can be highlighted by other screening approaches, in vitro assays and signals identified in the standard short term acute and chronic toxicity in vivo tests, which could be integrated into the process of identification of chemicals acting as non-genotoxic carcinogens. The use of omics technology, particularly transcriptomics, could be fundamental for the derivation of improved mechanistic understanding of the behavior of the tested chemical. Pathway-based toxicity does not only provide essential information to identify the chemical mode of action, but also the possibility of refining the point of departure to better to calculate the safe dose for human exposure, and the tipping point that may lead to the next adverse key event.

The results from the first attempts to develop omics-based CTA models show that most, if not all, key events and biological processes leading to oncotransformation are common to all three current models of the CTA, despite the differences in the grade of cell progression towards transformation. Gene transcripts enrichment for each process, however, marks the ability of each model to highlight different aspects of the process. Primary SHE cells allow the identification of several gene signatures related to cytoskeleton remodeling, the first necessary condition to pave the way to malignant changes, and the events related to cell-cycle control and senescence bypassing. Thus, the BALB/c 3T3 CTA proves to be an excellent model for investigating the role of inflammasome and immune-mediated inflammation in malignancy through the epithelial-mesenchymal transition, while the Bhas 42 CTA is the better model to investigate the mitogenic signals downstream of the activation of key oncogenes and with only the RAS gene activated. However, during transformation of the Bhas 42 cells, many cancer-associated signals are activated or repressed, not only signals down-stream of RAS gene activation.

Collectively, these results show that the CTA is ready to be included in the IATA for NGTxC, and that we can now address the key data gap identified for the NGTxC IATA as discussed in Jacobs et al. [3] (Figure 6).

However, ensuring the reproducibility of these results could be the key to making the CTA a more consistent and solid assay, sufficiently improving confidence for its application in the regulatory context, and this further work is a high priority. The development of a regulatory decision tree framework for specific targeted CTA use, on the basis of the information generated from the earlier IATA key events, within the IATA for NGTxC, is in progress.

## Figures and Tables

**Figure 1 ijms-24-05659-f001:**
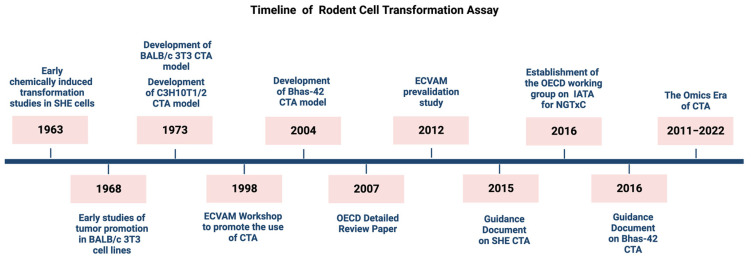
Milestones in the development of rodent cell transformation assay.

**Figure 2 ijms-24-05659-f002:**
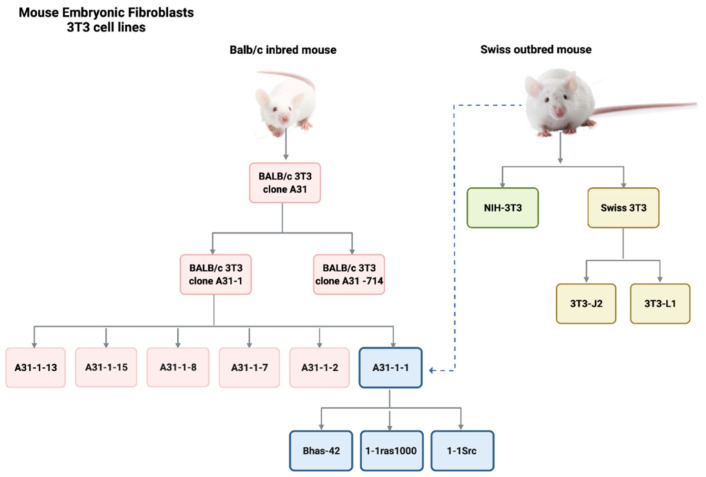
Family tree to show the origin of established 3T3 mouse embryonic fibroblasts.

**Figure 3 ijms-24-05659-f003:**
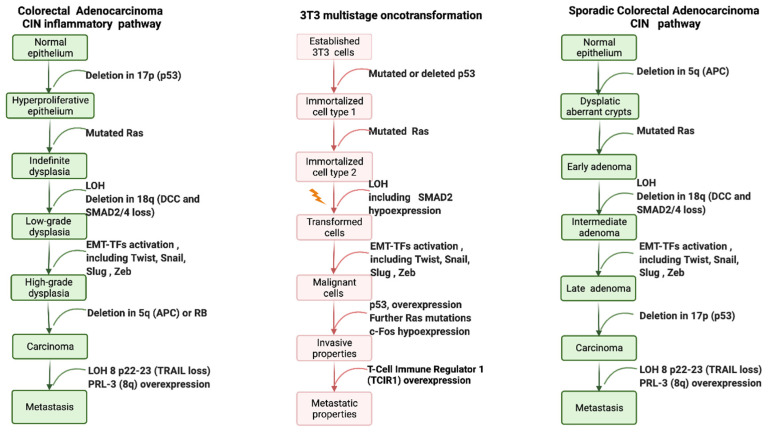
Schematic simplified representation of the key molecular changes required for acquiring a fully malignant phenotype in human cancer, and in vitro oncotransformation.

**Figure 4 ijms-24-05659-f004:**
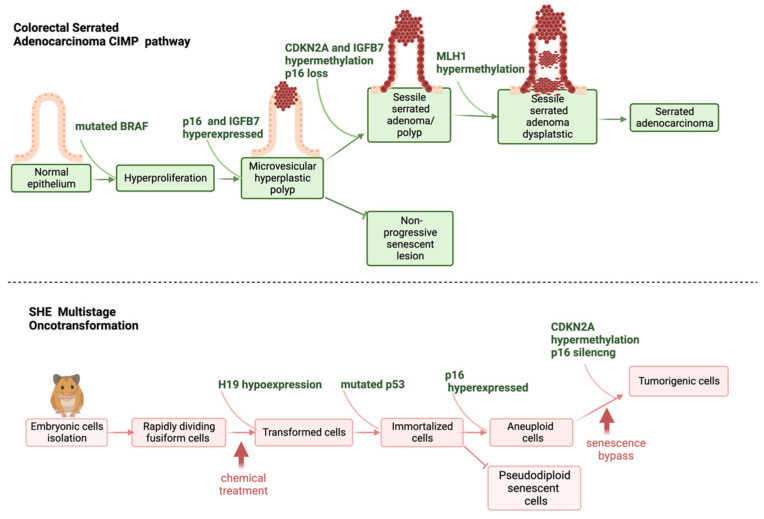
Schematic representation of steps leading to human serrated adenocarcinoma, and SHE cells oncotransformation.

**Figure 5 ijms-24-05659-f005:**
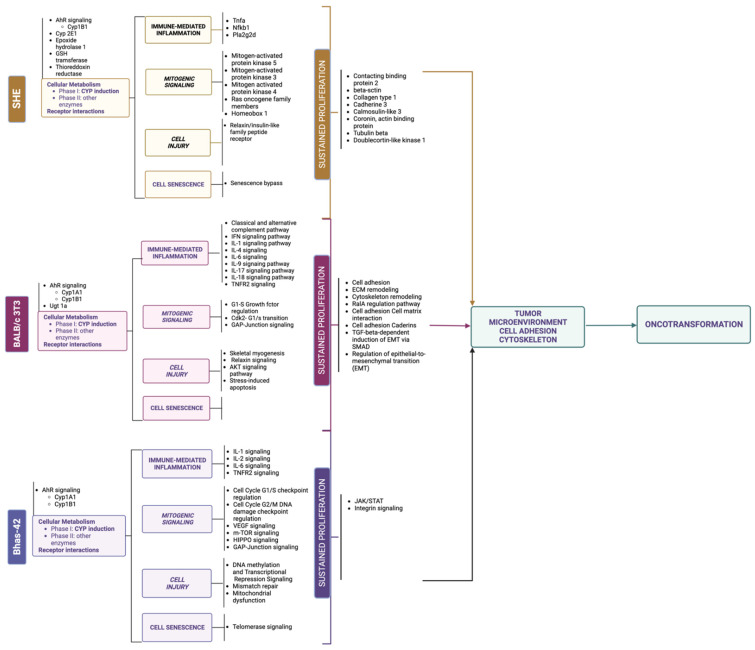
Priority biomarkers identified in transcriptomic prescreening approaches in the NGTxC IATA, biomarkers in the cell transformation assays.

**Figure 6 ijms-24-05659-f006:**
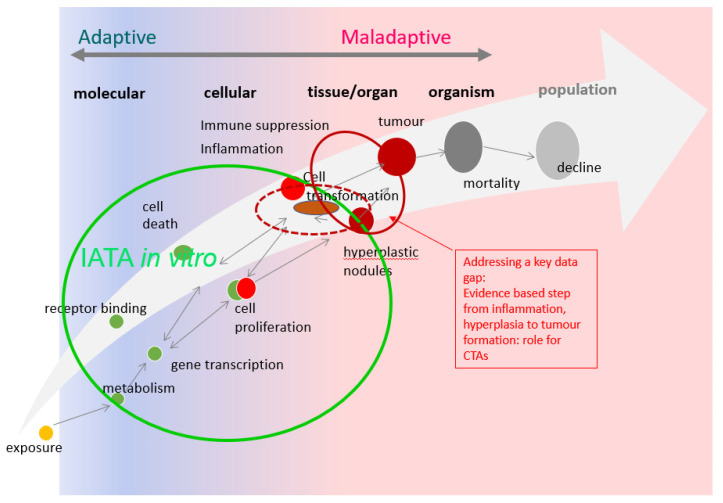
Conceptual overview of the adaptive versus maladaptive critical data gaps for adverse outcome recognition in NGTxC, and how they can now be overcome using the CTAs. From adaptive to maladaptive disease progression: key data gaps in the testing and assessment of non-genotoxic carcinogenicity (updated from [3,6], with authors/copyright holders permission). There are numerous in vitro assays to address the early key events from receptor binding and transactivation, gene transcription, metabolism and cell proliferation (indicated by the green circle on the left of the figure) [3]. The CTAs will be able to address the key data gap for cell transformation, both for early (initiation) and later (promotion) phases (broken red line ellipse). A change in morphology represents the point at which adaptive (sustained) proliferation and hyperplasia/dysplasia become maladaptive, the CTAs are the crucial tests to ensure an evidence based in vitro translation from the in vivo hyperplasia to tumor formation (solid red lined ellipse). This tipping point is histopathologically characterized with cellular and/or structural atypia, and this change is often observed as abnormal nuclear division and disorganized cell proliferation with loss of cell polarity, which the CTAs now show that they can address.

**Table 1 ijms-24-05659-t001:** Criteria for discriminating between genotoxic and non genotoxic chemicals according to the OECD CTA expert panel.

Tests Considered	Refined Categorisation
*In vitro tests* Bacteria reverse mutation test (TG 471) In vitro Mammalian Cell Gene Mutation test (TG 476) In vitro Mammalian Chromosome Aberration Test (TG 473) In vitro Micronucleus test (cell cultures) (TG 487) *In vivo tests* Rodent dominant lethal test (TG478) Mammalian spermatogonial chromosome aberration test (TG483) Mouse heritable translocation assay (TG485) Unscheduled DNA synthesis in mammalian liver cells (TG486) Mammalian erythrocyte micronucleus test (TG474) Mammalian bone-marrow chromosomal aberration test (TG475) Transgenic animal models DNA damage (alkaline Comet assay, TG489)	*Criteria for a non genotoxic chemical * Ames negative and In vitro cytogenetic studies, in vitro mammalian cell mutation studies (mouse lymphoma assay interpreted with new criteria) negative or Standard in vivo tests negative (An in vivo negative result overrules an in vitro positive result covering the same endpoint of genotoxicity *Criteria for a genotoxic chemical* Positive in vitro data that are confirmed in vivo

**Table 2 ijms-24-05659-t002:** Data Sources for updating information on animal and/or human carcinogenesis.

Data Source	Search Strategy	Links
International Agency for Research on Cancer	List of Classification of Agents from IARC Monographs 1–129 IARC Monographs	https://monographs.iarc.who.int/agents-classified-by-the-iarc/ (accessed on 26 July 2022)
U.S. National Toxicology Program	NTP Study Reports Collection Toxicology/Carcinogenesis 15th Report on Carcinogens (2021)	https://ntp.niehs.nih.gov/whatwestudy/testpgm/cartox/index.html (accessed on 2 November 2022) https://ntp.niehs.nih.gov/whatwestudy/assessments/cancer/roc/index.html (accessed on 2 November 2022)
Lhasa Ltd.	Carcinogenicity Database	https://carcdb.lhasalimited.org/study-information/44643857 (accessed on 26 July 2022)
National Library of Medicine	PubChem PubMed	https://pubchem.ncbi.nlm.nih.gov/compound/55784#section=Classification (accessed on 26 July 2022) https://pubmed.ncbi.nlm.nih.gov (accessed on 30 December 2022)
European Chemical Agency (ECHA)	ECHA C&L Inventory	https://echa.europa.eu/it/information-on-chemicals/cl-inventory-database (accessed on 30 December 2022)
World Health Organization (WHO)	Database of International Chemical Safety Cards (ICSCs)	https://www.ilo.org/safework/info/publications/WCMS_113134/lang--en/index.htm (accessed on 30 December 2022)
US Environmental Protection Agency	Integrated Risk Information System (IRIS)	https://www.epa.gov/iris (accessed on 26 July 2022)

**Table 3 ijms-24-05659-t003:** Carcinogenic chemicals (initially considered non-genotoxic) able to induce cell transformation in the SHE CTA, updated from reference [37]. See also Appendix A.

Testing Chemical								
CAS Registry Number	Chemical Name	Experimental Conditions Test at pH	Use	IARC Classification ^1^	CMR Classification	NTP (RoC) ^2^	Properties of Concern ^3^	Updated Information on Genotoxicity
88133-11-3	Bemitradine	6.7	Diuretic antihypertensive drug	NA	NA	Positive (NL)	NA	
94-36-0	Benzoyl peroxide (BPO)	7.0	Drug product	3	NA	Negative	Other concerns	
85-68-7	Butylbenzylphthalate	6.7	Plasticizer	3	R2	IE/(NL)	CLP notification: ED	Positive Chromosomal Aberration in vivo
105-60-2	Caprolactam	7.0	Used in nylon manufacture	3	NA	Negative	Other concerns, at pH of 7.3 was positive	Clear Negative in vivo
637-07-0	Clofibrate	pH 6.7 and pH 7.0	Antilipidemic and anticholesteremic drug	3	NA	NA	Other hazards	
117-81-7	di (2-Ethylhexyl) phthalate (DEHP)	pH 6.7 and pH 7.0	Plasticizer	2B	R1B	Reasonably anticipated to be carcinogenic (RoC)	CLP notification: ED	Positive in transgenic model
111-42-2	Diethanolamine	pH 6.7	Emulsifier	2B	NA	Positive in mouse both sexes, but not in rats (NL)	CLP notification: R2	
105-55-5	*N*,*N’*-Diethylthiourea	pH 6.7	Rubber chemical	3	NA	Positive in rats both sexes, but not in mouse (NL)	CLP notification: skin sensitizing	old data: Positive MLA
100-41-4	Ethyl benzene	pH 6.7	Intermediate	2B	NA	LE (NL)	CLP notification: C2	
72-43-5	Methoxychlor	pH 6.7	Pesticide	3	NA	Negative	No hazard	Positive MLA (old and new data), ND in vivo
93-15-2	Methyl eugenol	pH 6.7	Flavoring agent	2B	NA	Reasonably anticipated to be carcinogenic (RoC)	CLP notification: M2, C2	Positive evidence in transgenic model Positive comet in vivo
21340-68-1	Methylclofenapate	pH 7.0	Antilipidemic drug	NA	NA	NA	NA	
434-07-1	Oxymetholone	pH 6.7	Anabolic steroid	NA	NA	Reasonably anticipated to be carcinogenic (RoC)	No notification	
50-55-5	Reserpine	pH 6.7	Antihypertensive drug	3	NA	Reasonably anticipated to be carcinogenic (RoC)	Other hazards	Positive in vivo Micronucleus
16561-29-8	12-O-Tetradecanoyl phobol 13-acetate (TPA)	pH 6.7 and pH 7.0	Phorbol ester	NA	NA	Positive (NL)	Other hazards	
50892-23-4	Wyeth-14,643	pH 6.7	Pharmaceutical	NA	NA	Negative	CLP notification: C1B	Positive CA in vitro, Positive transgenic rodent

NA = not available; LE = Limited Evidence; IE = inadequate evidence; ED = Endocrine Disruptor; NL = Not listed in the primary reference RoC, but available from other sources such as the NTP website (https://ntp.niehs.nih.gov/whatwestudy/testpgm/cartox/index.htm, accessed on 2 November 2022l). ^1^ Reviewed according to 2019 update to the IARC Monographs Preamble [128]; ^2^ Updated according to 15th Report on Carcinogens [129]; ^3^ Properties of concerns according to hazard notifications to ECHA. Only notifications for cancer hazard and endocrine disruptor properties are reported.

**Table 4 ijms-24-05659-t004:** Non-genotoxic mechanisms and MOA of carcinogenic chemicals for which a clear correlation exists between transforming properties and in vivo carcinogenesis.

Group A Chemicals	Target Organs in Rodents	Initiation/Promotion In Vivo and/or In Vitro Tests	Epidemiological Evidence	Mechanisms
di (2-Ethylhexyl) phthalate (DEHP)	Liver, hepatocarcinoma (RM, RF, MM, MF) Liver (PPAR-null transgenic mice) Benign testicular tumors (RM) Benign pancreatic tumors (RM) [129]	Acting as a promoter in two different strains of mice	An association with breast cancer has recently been reported [130]	Activation of PPARα in rodents. Alternative mechanisms in PPAR-null transgenic mice may include CAR activation and peroxisome proliferation. None of these mechanisms were confirmed in humanized PPAR-null transgenic mice [131]. Possible implication of AhR-mediated signaling and activation of CYP 1B1 [132]. Antiandrogenic activity (identified on list 1 as an Endocrine Disruptor at the EU level) and possible implication of epigenetic DNA methylation disruption properties of DEHP, with links with cancer requiring to be elucidated [74]
Diethanolamine (DEA)	Liver neoplasms (hepatocarcinoma and hepatoblastoma) (MM, MF) [133]		No data	Perturbation of choline homeostasis [134]
*N*,*N*′-Diethylthiourea (DETU)	Carcinoma of Thyroid Gland Follicular Cells (RM, RF) [135]		No data	Possible inhibition of thyroid hormone biosynthetic enzyme by the thiourea/thiocarbonyl moiety via formation of disulfide bridge [136]. DETU also affects the metabolic profile of cholesterol (increase), arachidonic acid (decrease), long chain carnitine contents (decrease) [137]
Ethylbenzene	Kidney neoplasms (RM) (inhalation) Testis neoplasms (RM) (inhalation) [138,139]		No data	Associated with rat nephropathy, following an accumulation of α2u-globulin [140]. This mechanism is considered not human relevant [141]
Methyl eugenol	Liver adenomas and carcinomas (RM, RF MM, MF) Stomach neuroendocrine tumors (RM, RF, MM) [129]		No data	Oxidative stress [142,143]
Oxymetholone	Liver adenomas and carcinomas (RF) [129]		Limited evidence of leukemia, liver cancer, or esophageal cancer following oxymetholone treatment. A case of ampullary carcinoma (bile-duct) has been also reported [129] A case of hepatocarcinoma in a AAS abuser has been described [144]	Modulation of androgen receptor [145]
Reserpine	Mammary gland neoplasms (MF) Seminal vesicles undifferentiated carcinoma (MM) Adrenal gland pheocromocytoma (RM) [129]	Acting as a promoter in rats	Increased risk of breast cancer among individuals who had used reserpine for over 10 years [146] Small but significant increase in the risk of breast cancer with reserpine. This finding was not confirmed by prospective studies [147]	Chromaffin cell proliferation is the postulated mechanism for pheocromocytoma [148]. Increase of serum levels of prolactin could be responsible for mammary gland tumors [149]. Both mechanisms are related to the ability of reserpine to affect the neural response
**Group B**	**Target Organs in Rodents**	**Initiation/Promotion In Vivo and/or In Vitro Tests**	**Epidemiological Evidence**	**Mechanisms**
Wyeth-14,643	Liver adenomas and carcinomas (RM, MM) [150,151,152,153,154] No neoplastic formation in p53^+*/*−^ mice [155]		No data	PPARα dependent peroxisome proliferation
**Group C**	**Target Organs in Rodents**	**Initiation/Promotion In Vivo and/or In Vitro Tests**	**Epidemiological Evidence**	**Mechanisms**
Bemitradine (BEM)	Liver adenoma and carcinoma (RM) Mammary neoplasms (RF) [156]	Positive in rat altered hepatic foci model [156]		CAR-activator and related increase of Cyp3b1 [157]
Butylbenzyl phthalate (BBP)	Pancreas: Adenoma- acinar cell [158]	Increased incidence of prostate intraepithelial neoplasm in rats treated with 3,2′-dimethyl-4-aminobiphenol (DMAB) [159]	No significant association with breast cancer risk in occupationally exposed women [160,161]	A PPARα-dependent mode of action (has been proposed for the induction of pancreatic acinar cell tumors [162] BBP can also act as a weak AhR agonist and modulate AhR-mediated signaling pathway [132,163]
Clofibrate	Liver adenomas and carcinomas [164] Pancreas carcinoma acinar cells [165] No neoplastic formation in p53^+*/*−^ mice. Non-neoplastic findings in the adrenals, pancreas, and prostate [155].		Patients given clofibrate developed several adverse health conditions but not cancer.	PPARα dependent peroxisome proliferation
Methylclofenapate	Liver hepatocarcinoma (RM, RF, MM, MF) Pancreas adenoma (RM) Testes adenoma Leyding cells (RM) [166,167]			
12-O-Tetradecanoyl phobol 13-acetate (TPA)	Potent promoter of the skin carcinogenesis in mouse Complete carcinogen	Used as a promoter in in vitro and in vivo initiation-promotion tests TPA alone induces a significant incidence (*p* value 0.05) of papillomas and some carcinomas in mouse skin [95]	No data	TPA mimics the action of diacylglycerol (DAG), thus activating several receptors downstream of the signaling pathway
**Group D**	**Target Organs in Rodents**	**Initiation/Promotion In Vivo and/or In Vitro Tests**	**Epidemiological Evidence**	**Mechanisms**
Benzoyl peroxide (BPO)	Negative results	Acts as a promoter in mouse skin initiated with dimethylbenz(a)anthracene (DMBA) [168]	No data	As a free radical generating chemical, it was found to induce direct oxidative activation of protein kinase C [169,170]
Methoxychlor	Liver: hemagiosarcomas (RM)		An association between leukemia and farmers using methoxychlor has been reported (OR 2.2) [171]	Methoxychlor metabolites interact with estrogen and androgen receptors [172]

Abbreviations: MF, mouse female; MM, mouse male; RF, rat female; RM, rat male.

**Table 5 ijms-24-05659-t005:** List of non-genotoxic carcinogenic chemicals able to induce cell transformation in the BALB/c 3T3 CTA.

Test Chemical				Carcinogenicity Evidence			
CAS	Chemical Name	Clone/Experimental Protocol	Use	IARC Classification ^1^	CMR Classification	NTP (RoC) ^2^	Properties of Concern ^3^
Registry Number							
87-29-6	Cinnamyl anthranilate	A31-1	Flavoring substance	3	NC	Positive (NL)	No hazards
56-53-1	Diethylstilbestrol (DES)	A31	Synthetic nonsteroidal estrogen. Former use: miscarriage prevention, hormone replacement therapy. Current use drug in clinical trials for the treatment of prostate and breast cancer			Known to be a human carcinogen	
						First listed in the *First Annual Report on Carcinogens* [129]	
50-29-3	Dichlorodiphenyl trichloroethane (DDT)	A31	Pesticide	2A	NA	Reasonably anticipated to be a human carcinogen [129]	CLP notification: C2
							ED
50-28-2	Estradiol	A31-1-13	Hormone replacement therapy; combined oral contraceptives	NA	R 1A	Known to be human carcinogen [129]	CLP notification: C2
64-17-5	Ethyl Alcohol Ethanol	A31-1-13	Industrial use in the manufacture of drugs, plastics, lacquers, polishes, plasticizers, and cosmetics; medical uses as a topical anti-infective, and as an antidote for ethylene glycol or methanol overdose. Commercial use in beverages	1	NC	NA	CLP notification: C 1B
598-55-0	Methylcarbamate	A31-1-1	Primarily used in the textile and polymer industries as a reactive intermediate	3	NA	Clear evidence in rats both sexes, negative in mouse (NL)	CLP notification: C2
34807-41-5	Mezerein	A31-1-13	Daphnetoxin, folk medicine plant used in cancer treatment	NA	NA	NA	Other hazards
139-13-9	Nitrilotriacetic acid, trisodium salt	A31-1-13	Boiler feedwater additive, in water and textile treatment, in metal plating and cleaning and in pulp and paper processing	2B	NA	Reasonably anticipated to be a human carcinogen [129]	CLP notification: C2
50-55-5	Reserpine	A31-1-1	Antihypertensive drug	3	NA	Reasonably anticipated to be carcinogenic [129]	Other hazards
128-44-9	Sodium saccharin	A31-1-13	Artificial sweetener	NA	NA	Delisted from RoC [129]	NA

NA = not available; NC = Not Classified; NL = not listed; LE = Limited Evidence; IE = inadequate evidence; ED = Endocrine Disruptor. ^1^ Reviewed according to 2019 update to the IARC Monographs Preamble [128]. ^2^ Updated according to 15th Report on Carcinogens [129]. ^3^ Properties of concern according to hazard notifications to ECHA. Only notifications for cancer hazard and endocrine disruptor properties are reported.

**Table 6 ijms-24-05659-t006:** Overall correlation between transforming properties and in vivo carcinogenesis.

SHE CTA Transforming Chemicals	BALB/c 3T3 Transforming Chemicals
Group A: Chemicals for which a clear correlation exists between transforming properties and in vivo carcinogenesis	
Di (2-Ethylhexyl) phthalate (DEHP)	Di (2-Ethylhexyl) phthalate (DEHP)
Reserpine	Reserpine
Diethanolamine	Diethylstilbestrol (DES)
*N*,*N*′-Diethylthiourea (DETU)	Dichlorodiphenyl trichloroethane (DDT)
Ethylbenzene	Estradiol
Methyl eugenol	Nitrilotriacetic acid, trisodium salt
Oxymetholone	Ethanol
Group B: Chemicals for which available data indicate a possible correlation between transforming properties and in vivo carcinogenesis	
Wyeth-14,643	Methylcarbamate
Group C: Chemicals for which data are suggestive for a possible correlation between transforming properties and in vivo carcinogenesis	
Bemitradine	Cinnamyl anthranilate
Butylbenzoylphthalate	
Clofibrate	
Methylclofenapate	
12-O-Tetradecanoyl phobol 13-acetate (TPA)	
Group D: Chemicals for which not enough data are available to show a correlation between transforming properties and in vivo carcinogenesis	
Benzoyl peroxide	Mezerein
Methoxychlor	

**Table 7 ijms-24-05659-t007:** Non-genotoxic mechanisms and MOA of carcinogenic chemicals for which a clear correlation exists between transforming properties and in vivo carcinogenesis in BALB/c 3T3 cells.

Group A Chemicals	Target Organs in Rodents	Initiation/Promotion In Vivo and/or In Vitro Tests	Epidemiological Evidence	Mechanisms
Diethylstilbestrol (DES)	Several different tissue sites (primarily estrogen-sensitive organs and tissues). Such as: mammary gland, carcinoma, adenocarcinoma (MM, MF) cervix and uterus, adenocarcinoma, vagina (squamous-cell-carcinoma (MF)) [129]	No data	Sufficient evidence of carcinogenicity from studies in humans.	As a synthetic estrogen, DES can react with the estrogen receptors, but the molecular mechanisms triggering and sustaining the pathway to cancer are still poorly understood [173]
Di (2-Ethylhexyl) phthalate (DEHP)	See Table 5	See Table 5	See Table 5	See Table 5
Dichlorodiphenyl trichloroethane (DDT)	Liver, hepatocellular (MM, MF, R)	Positive in CTA BALB/c mouse embryo [175,176]	Epidemiological studies gave mixed results, showing positive associations with breast cancer in women subgroups exposed to high levels of DDT, multiple myeloma in farmers, and liver cancer in high-level exposed population. Negative associations were also reported	DDT behaves as an estrogen receptor agonist and/or androgen receptor antagonist. Due to their long half-life and its lipophilic nature, DDT and its metabolite DDE (dichlorodiphenyl dichloroethylene) are still detected in the serum of western pregnant and lactating mothers [177,178]
	Multiple sites, sarcoma-reticulum cell (MF) [174]			
Estrogen	Endometrial, cervical, and mammary-gland tumors in mice, mammary and pituitary-gland tumors in rats, and kidney tumors in hamsters [129]	Positive in CTA BALB/c A31-1-13	Increased incidence of endometrial and ovarian cancer as a consequence of the therapy [129]	Estrogen receptor interaction causing cell proliferation, affecting cell differentiation and gene expression [129]. AhR interaction leading to Cyp1A1 and Cyp1B1 mediated modulation of estrogen metabolism [10,67,179]
Nitrilotriacetic acid (NTA), trisodium salt	Kidney: adenocarinoma–tubular (RM, RF, MM, MF) [129];	Positive in CTA BALB/c mouse embryo cells [180]	Data available from epidemiological studies are inadequate [129]	It has been suggested that the nephron-carcinogenic properties of NTA are related to dose-dependent changes in intracellular zinc ion homeostasis, due to the chelating properties of NTA
	Urinary bladder: Carcinoma-squamous cell; Carcinoma-transitional cell (RF)			
	Ureter: Adenoma-papillary; Papilloma (RM) [129]			
Reserpine	See Table 6	See Table 6	See Table 6	See Table 6

**Table 8 ijms-24-05659-t008:** List of non-genotoxic carcinogenic chemicals able to induce cell transformation in the Bhas 42 CTA. (Note that these data have not undergone the criteria evaluation as conducted for Table 1, Table 2, Table 3, Table 4, Table 5, Table 6 and Table 7, which were included in the OECD discussions of 2014).

CAS Registry Number	Chemical Name	Carcinogenicity	Genotoxicity Studies with Negative or Equivocal Results
10108-64-2	Cadmium chloride	IARC class 1	Ames, in vitro CA
120-80-9	Catechol	IARC class 2B	Ames, in vivo MN
474-25-9	Chenodeoxicholic acid	Colon cancer promoter	Ames
3165-93-3	4-Chloro-o-toluidine hydrochloride	IARC class 2A	Ames
81-25-4	Cholic acid	Colon cancer promoter	Ames
83-44-3	Deoxycholic acid	Colon cancer promoter	Ames
50-29-3	DDT	IARC class 2A	Ames in vitro CA, in vivo MN
111-42-2	Diethanolamine	IARC class 2B	Ames, in vitro CA in vivo MN
117-81-7	DEHP	IARC class 2B	Ames, in vivo MN, in vitro MN, MLA
116355-83-0	Fumonisin B1	IARC class 2B	Ames
5989-27-5	D-Limonene	IARC class 3, Male rat kidney tumors	Ames, in vivo comet, in vitro CA
434-13-9	Lithocholic acid	Colon cancer promoter	Ames
135-23-9	Methapyrilene HCl	Hepatocarcinogen in rats	Ames, in vitro CA, SCE, in vivo CA, in vivo MN
124-58-3	Methylarsonic acid	IARC class 2B	Ames
34807-41-5	Mezerein	Tumor promoter on mouse skin	Ames
78111-17-8	Okadaic acid	Tumor promoter on mouse skin	Ames, in vitro CHO/HGPRT
57-83-0	Progesterone	IARC class 2B	Ames in vitro CA, SCE, in vivo CA
16561-29-8	12-O-Tetradecanoyl phobol 13-acetate (TPA)	Tumor promoter on mouse skin, Carcinogen on heirless mouse skin, Carcinogen in NTP	Ames
7631-89-2	Sodium arsenate	IARC class 1	Ames, MLA
7784-46-5	Sodium arsenite	IARC class 1	Ames
82385-42-0	Sodium saccharin	IARC class 3, Rat and mouse bladder tumors	Ames, in vivo CA, in vivo comet, in vitro MLA
1746-01-6	2,3,7,8-Tetrachlorodibenzo-P-dioxin (TCDD)	IARC class 1	Ames, in vitro MLA, in vitro CA, SCE, in vivo CA

Non genotoxic = Ames-negative or Ames-discordant. Reviewed according to 2021 update to the IARC Monographs Preamble. Ames = reverse bacterial mutation assay in Salmonella typhimurium; in vitro CA = in vitro chromosomal aberration assay; in vivo MN = micronucleus test; SCE = sister chromatid exchange; MLA = mouse lymphoma assay; CHO = Chinese hamster ovary; CHL = Chinese hamster lung; HGPRT = hypoxanthine guanine phosphoribosyltransferase. Note that these data have not undergone the criteria evaluation as conducted for Table 1, Table 2, Table 3, Table 4, Table 5, Table 6 and Table 7, as were included in the OECD discussions of 2014.

**Table 9 ijms-24-05659-t009:** BALB/3T3 cells, clone A31-1-1 cell growth in different culture conditions using FBS-supplemented medium or synthetic serum-supplemented medium ^1^.

	Degree of Cell Confluence at Different Time after Cell Seeding											
Cell Seeding Density	24 h			48 h			72 h			96 h		
	C	MIX	XF	C	MIX	XF	C	MIX	XF	C	MIX	XF
5 × 10^5^	50	50	20	70–80	60–70	20	>90	80–90	10	/	/	10
2.5 × 10^5^	20–30	30	10–20	50–60	50	10	80	50–60	10	/	/	10
1 × 10^5^	10	10	<10	40	10–20	<10	50–60	10–15	<10	>90	10–15	<10
0.5 × 10^5^	<10	<10	ND	20	10	<10	30–40	10	<10	70–80	10	<10

^1 ^Minimum essential medium Eagle (MEM) has been used for culturing BALB/3T3 cells, clone A31-1-1; XerumFree^TM^ (XF) has been obtained from TNCBIO and certified as human and animal component free. C = 10% FBS-supplemented MEM; MIX = mixture 25% C: 75% XF; XF = 2% XF-supplemented MEM.

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
