# Peer review of "The Cell Transformation Assay: A Historical Assessment of Current Knowledge of Applications in an Integrated Approach to Testing and Assessment for Non-Genotoxic Carcinogens"

_ijms, 2023, doi:10.3390/ijms24065659_

Round 1

Reviewer 1 Report

It's excellent draft to understand readers interested in thie fleld.

Author Response

We appreciate the positive feedback from the reviewer

Reviewer 2 Report

In this review, the authors discuss the latest research trend on application in an integrated approach to testing and assessment for non-genotoxic carcinogens. This review presents highly significant strategies for the risk assessment of non-genotoxic carcinogens based on a vast amount of data. However, the paper's readability is compromised by excessively long and occasionally grammatically incorrect sentences, although this is not true for the entire paper. I think that this manuscript need to be revised by a professional linguistic reviewer before considering the manuscript for publication.

The following examples illustrate the difficulty in accurately understanding the intended meaning.

For example;

Application of this knowledge is utilised to address how the different types of CTAs variously addressing initiation and promotion, can be included on a mechanistic basis within the integrated approach to testing and assessment for non-genotoxic carcinogens.

Building upon assay assessments targeting the key events in the IATA, we identify how the different CTA models can appropriately fit, following prescreening transcriptomic approaches, and assessment within the earlier key events of inflammation, immune disruption, mitotic signaling and cell injury and later key events of (sustained) proliferation and change in morphology leading to tumor formation.

Author Response

We want to thank the reviewer for his comments and suggestions.
As suggested, we have carefully reviewed the entire manuscript to improve the paper's readability, as shown in the revised manuscript..  A complete review of the Engish style has been performed by the English mother-tongue co-author.

Reviewer 3 Report

In my opinion this review is really illuminating for researchers like me who study environmental pollutants and carcinogenesis. The authors have dealt with the subject in a clear and exhaustive manner by inserting clear and pertinent figures and tables. It is a complete review in all its parts also in reporting the history of the somatic mutation theory of carcinogenesis. The different sections of the review (introduction, discussion and conclusions) are in perfect balance with each other. The references are correct and sufficient.

Author Response

We are appreciative of the reviewer's positive feedback.